# Phosphatase protector alpha4 (α4) is involved in adipocyte maintenance and mitochondrial homeostasis through regulation of insulin signaling

Masaji Sakaguchi [1] ✉, Shota Okagawa[1], Yuma Okubo[1], Yuri Otsuka[1], Kazuki Fukuda[1,2], Motoyuki Igata[1], Tatsuya Kondo[1], Yoshifumi Sato [3], Tatsuya Yoshizawa [3], Takaichi Fukuda[4], Kazuya Yamagata[2,3], Weikang Cai[5], Yu-Hua Tseng [6], Nobuo Sakaguchi[7,8], C. Ronald Kahn [6] & Eiichi Araki [1,2]

Insulin signaling is mediated via a network of protein phosphorylation. Dysregulation of this network is central to obesity, type 2 diabetes and metabolic syndrome. Here we investigate the role of phosphatase binding protein Alpha4 (α4) that is essential for the serine/threonine protein phosphatase 2A (PP2A) in insulin action/resistance in adipocytes. Unexpectedly, adipocyte-specific inactivation of α4 impairs insulin-induced Akt-mediated serine/threonine phosphorylation despite a decrease in the protein phosphatase 2A (PP2A) levels. Interestingly, loss of α4 also reduces insulin-induced insulin receptor tyrosine phosphorylation. This occurs through decreased association of α4 with Y-box protein 1, resulting in the enhancement of the tyrosine phosphatase protein tyrosine phosphatase 1B (PTP1B) expression. Moreover, adipocyte-specific knockout of α4 in male mice results in impaired adipogenesis and altered mitochondrial oxidation leading to increased inflammation, systemic insulin resistance, hepatosteatosis, islet hyperplasia, and impaired thermogenesis. Thus, the α4 /Y-box protein 1(YBX1)-mediated pathway of insulin receptor signaling is involved in maintaining insulin sensitivity, normal adipose tissue homeostasis and systemic metabolism.

Obesity, especially central obesity, results in whole-body insulin resistance and predisposes to development of a range of metabolic diseases, including type 2 diabetes mellitus, nonalcoholic fatty liver diseases (NAFLD), atherosclerotic cardiovascular diseases, and cancer[1,2]. Adipose tissue is functionally heterogeneous with white adipose tissues (WAT) being mainly responsible for storing energy as triglycerides (TGs), while beige and brown adipose tissues (BAT) burn energy through their high rates of mitochondrial respiration due to the presence of uncoupling protein 1 (UCP1)[3]. Several hormones, and particularly insulin, serve as important regulators of this energy balance. Indeed, insulin is an important regulator of brown and white adipogenesis and adipocyte survival[4] and promotes energy storage by inhibiting lipolysis and promoting glucose uptake and its conversion into lipids[2]. Conversely, when caloric supply is limited, lower circulating insulin levels lead to disinhibition of lipolysis and mobilization of lipids to meet energy demand. Thus insulin sensitivity of adipose tissues plays a critical role in normal metabolic homeostasis[5].

Insulin resistance is a central feature of many metabolic diseases. Insulin resistance can be the result of several pathogenic factors, including increases in inflammatory cytokines, altered levels of free

fatty acids and other circulating metabolites, and ectopic lipid accumulation, all of which can lead to increases in inhibitory Ser/Thr phosphorylation of the insulin receptor and its substrates, changes in mitochondrial function and activation of the unfolded protein response (UPR) and oxidative stress[6–10]. In obesity, enhanced hydrolysis of adipocyte triglycerides (TGs) results in increased circulating levels of free fatty acids (FAs), diacylglycerols (DAGs), small dense low-density lipoproteins (sdLDLs) particles and ceramides, all of which may contribute to insulin resistance[11,12]. Increased visceral adipose tissues is also accompanied by the production of proinflammatory cytokines, such as TNFα, IL-1β, and IL-6, which induce monocyte infiltration and adipocyte apoptosis[13,14]. These inflammatory events can result in reduced insulin signaling in adipose tissue and other tissues via multiple mechanisms, including activation of C-Jun N-terminal kinase and TLR-4, and NLRP3 inflammasome-mediated IL-1β secretion[15–17]. These pathological pathways are often intertwined, highlighting the complex causes and effects of impaired insulin response in metabolic diseases.

The insulin receptor (IR) is a tetrameric membrane Tyr kinase composed of two extracellular α-subunits and two transmembrane β−subunits harboring the Tyr kinase domain[2]. Upon ligand binding, IR undergoes a conformational change, which induces the Tyr phosphorylation of its β-subunits, resulting in the recruitment of protein substrates, such as insulin receptor substrate-1 and -2 (IRS-1 and IRS-2) and Shc[18]. This triggers a cascade of reactions leading to the activation of phosphatidylinositol 3 kinase (PI3-K) and Ras/MAPK signaling[19]. Downstream signals of PI3-K are mediated by multiple Ser/Thr kinases, including phosphoinositide-dependent protein kinase 1 (PDK-1), Akt, PKC, S6K, and GSK3, which regulate the activity of various transcriptional factors, such as FoxO1, PGC1α, SREBPs, and FoxK1[20–23]. These promote the metabolic actions of insulin on glucose, lipid, and mitochondrial metabolism, as well as effects on cell proliferation, differentiation, and apoptosis [reviewed in ref. 8].

Less is known about how these signals are turned off. When insulin levels fall, the insulin signaling cascade is reversed by the action of protein phosphatases, including phosphotyrosine phosphatases, such as protein Tyr phosphatase 1B (PTP1B), LAR, PTPα and others[24,25], which can dephosphorylate tyrosine residues of the insulin receptor and its substrates, and Ser/Thr protein phosphatases (e.g., PP2, PP4, and PP6)[26–30]. In comparison with the numerous classes of protein kinases involved in insulin action, including Akt, ERK, mTOR and the S6 Kinases, there are a relatively small number of Ser/Thr phosphatases. They have highly conserved catalytic subunits and, in general, broad and overlapping substrate specificity[29]. PP2A, which catalyzes the dephosphorylation of the Ser/Thr residues of the majority of the intracellular signaling molecules, has been shown to form a core dimer of consisting of a catalytic (C) subunit and a 65 kDa scaffolding subunit A, which then further associates with any of more than 23 regulatory B subunits. The B subunits likely determine the substrate specificity and subcellular localization, exerting multiple physiological functions. PP2A holoenzyme activity and specificity are controlled by a coordination of five PP2A modulators[31]. One of essential phosphatase regulator, Alpha4 (α4) was initially discovered as a PP2Ac binding protein that regulates mTOR activity in the yeast and mouse[26,32]. Alpha4 is associated with PP2Ac independently of the A or B subunits forming the noncanonical PP2A complex, which might eventually address to the unique target molecules. Alpha4 is thought to play a role to protect the phosphatases from the ubiquitin-dependent degradation and maintain the noncanonical PP2A[26,28]. While deletion of Alphah4 has shown increased basal phosphorylation levels of several PP2A targets, such as S6K[28,33], the role of Alpha4 in insulin action and adipose tissue remains unclear.

Here, we demonstrate a role of α4 in maintenance and differentiation of adipose tissue in vivo. Thus, we find that the protein phosphatase protector Alpha4 binds with Y-box protein 1 (YBX1), a transcription factor for Tyr phosphatase PTP1B gene, thus regulating PTP1B expression and subsequently controlling IR Tyr phosphorylation. This leads to impairment of mitochondrial function and energy expenditure, hyperglycemia with systemic insulin resistance, glucose intolerance and hepatosteatosis. The results demonstrate an important role of Alpha4 via, interacting with YBX1, in a feedback regulatory loop for IR Tyr phosphorylation in adipose tissue function.

# Results

## α4 regulates insulin signaling and adipogenesis

Assessment of the expression of *α4* mRNA in three adipose tissue depots revealed the highest levels in BAT, followed by inguinal WAT (iWAT) and then epididymal WAT (eWAT) (Supplementary Fig. 1a). To determine if α4 plays a significant role in BAT, we examined changes in expression of α4 during brown adipocyte differentiation using WT-1 immortalized murine brown preadipocyte[34] (Supplementary Fig. 1b). This revealed a stepwise increase in α4 mRNA and protein expression during the six-day culture of brown preadipocyte differentiation which correlated with the accumulation of lipid as seen by Oil Red O staining (Fig. 1a, b and Supplementary Fig. 1b, c). To assess the specific role of α4 in adipogenesis, we stably overexpressed α4 (α4OE) in WT-1 cells by using the lentivirus vector (pLV) (Fig. 1c). α4OE did not exert any effect on the differentiation as assessed by Oil Red O staining or expression of *Adiponectin*, *PPARγ*, or *Glut4*. Interestingly, however, α4OE cells showed increases in the levels of mitochondrial respiratory chain complexes, as well as the thermogenic marker *UCP1*, after adipogenic induction (Fig. 1d and Supplementary Fig. 1d–f). Additionally, in human brown preadipocytes in culture[35], α4OE also enhanced *UCP1* expression upon differentiation (Supplementary Fig. 1g–i). Furthermore, overexpression of α4 resulted in increased mitochondrial size as visualized by electron microscopy (Supplementary Fig. 1j). Functionally, the α4OE murine cells showed significant increases in the maximum oxygen consumption rate in basal respiration (36.7%), spare respiratory capacity (44.3%), and a trend toward higher levels of basal respiration and ATP production compared with the Control when using pyruvate/glucose as substrates (Fig. 1e, f). Conversely, knockdown of α4 in WT-1 using shRNA (α4KD) (Supplementary Fig. 1k) did not cause any detectable change in the cell growth, proliferation and DNA synthesis but resulted in marked reduction in expression of differentiation markers, such as *Adiponectin*, *PPARγ*, *Glut4*, *Adrb*, *AP2* and *UCP1*, and this was associated with decreased differentiation of preadipocytes as assessed by Oil Red O staining (Fig. 1g and Supplementary Fig. 1l–n).

α4 is an essential regulator of phosphatase PP2A activity by interacting with catalytic (C) subunits, protecting them from proteasomal degradation until they are assembled into a functional phosphatase complex[28]. PP2A has many substrates and potential to decrease Ser/Thr phosphorylation of insulin signaling molecules. Somewhat surprisingly, knockdown of α4 in brown preadipocytes resulted in a 50% decrease in IR (Y1162/1163) Tyr phosphorylation and a parallel decrease in Akt (S473) phosphorylation, but had no significant effect on insulin-stimulated ERK1/2 (T202/Y204) phosphorylation (Fig. 1h–k and Supplementary Fig. 1o, p). α4KD also enhanced ribosomal S6 protein (S235/S236) phosphorylation in the basal state, similar to its effect in mouse embryo fibroblasts[28] (Fig. 1h, l and Supplementary Fig. 1o). While PP2A is exclusively a Ser/Thr phosphatase[29], these results suggest that α4 may indirectly regulate both tyrosine and serine phosphorylation in the proximal insulin signaling cascade to regulate adipogenesis.

## α4 regulates IR Tyr phosphorylation via the YBX1-PTP1B axis

To investigate how α4 regulates the IR tyrosine phosphorylation, we conducted analysis of proteins that interact with α4 by using WT-1 cells transfected with 3XFlag-tagged α4 protein, followed by immunoprecipitation of associated proteins using anti-Flag beads and mass spectroscopic proteomic analysis (Fig. 2a and Supplementary

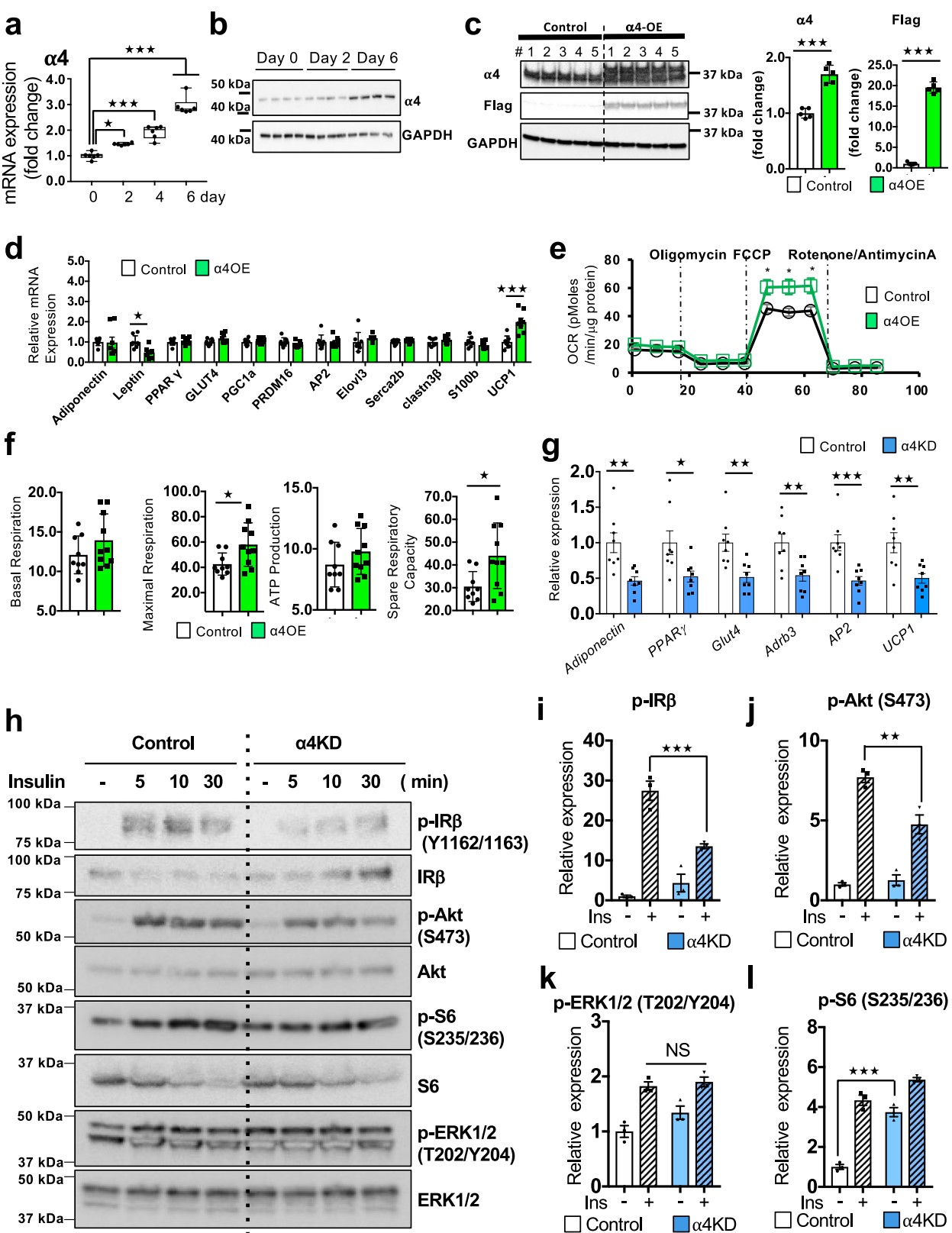

Fig. 2a, b). As expected, we identified PP2A and several other proteins known to associate with αc4 (Fig. 2b and Supplementary Fig. 2b, c). One unexpected interacting protein identified in these complexes was YBX1, a transcription factor, which has been previously shown to regulate expression of PTP1B[36] (Fig. 2b and Supplementary Fig. 2c). Although the proteomic analysis also detected other proteins,

further verification for interaction will be needed for the identified α4 interaction partners that were not validated in this study.

To examine the effect of α4/YBX1 complex on insulin signaling, primary brown preadipocytes isolated from α4[flox/flox] mice[33] were infected with adenovirus carrying Cre recombinase or GFP as control (Ad-Cre recombinase or Ad-GFP) (Fig. 2c). As observed in α4KD WT-1

**Fig. 1 | Loss of α4 in brown preadipocytes impaired insulin signaling and adipogenesis. a** Abundance of *α4* mRNA in mouse brown preadipocyte cells during the differentiation process before or on days 2, 4, and 6 after the induction of differentiation. Box plots are defined in terms of minima and maxima by whiskers, and the center and bounds of box by quartiles (one-way ANOVA post hoc Bonferroni test, $n = 6$ technical replicates per group, *$p = 0.01$, ***$p < 0.0001$). **b** Immunoblotting analysis of α4 in lysates from brown preadipocyte cells during the differentiation process before or 2 and 6 days after induction of the differentiation. ($n = 4$). **c** Immunoblotting for α4 and Flag in lysates from Control mouse brown preadipocytes or cells overexpressed 3XFlag-α4 (α4-OE). Densitometric analysis of α4 and Flag in mouse brown preadipocytes. Data are mean ± SEM of $n = 5$ (Two-tailed Student's *t*-test, ***$p < 0.0001$). **d** Adipocyte markers in differentiated brown adipocytes from Control ($n = 8$) and 3XFlag-α4 transfected (α4-OE) cells ($n = 8$). Data are mean ± SEM (Two-tailed Student's *t*-test, *Leptin*: *$p = 0.02$, *UCP1*: ***$p = 0.0004$). **e** Mitochondrial oxidative phosphorylation activity in mouse differentiated brown adipocytes from Control ($n = 9$) and α4-OE ($n = 10$) cells. Data are mean ± SEM (Two-tailed Student's *t*-test, *$p < 0.05$). **f** Quantitation of basal

respiration (*$p = 0.02$), maximal respiration capacity, ATP production and Spare Respiratory Capacity (*$p = 0.01$). Data are presented as mean ± SEM (Two-tailed Student's *t*-test, Control ($n = 9$), α4-OE ($n = 10$)). **g** Levels of adipocyte markers in differentiated brown adipocytes from Control cells and cells subjected to knockdown of α4 by shRNA (α4 KD) ($n = 8$ biologically independent cell clones per group). Data are mean ± SEM (Two-tailed Student's *t*-test, *Adiponectin*: **$p = 0.003$, *PPARγ*: *$p = 0.02$, *Glut4*: *$p = 0.003$, *Adrb3*: **$p = 0.07$, *AP2*: ***$p = 0.0009$, *UCP1*: **$p = 0.006$). **h** Immunoblotting analysis results showing insulin-dependent signaling molecules for IR, Akt, ERK, and S6 phosphorylation in lysates from α4 KD-treated or Control adipocytes after 100 nM insulin stimulation for the indicated durations. **i–l** Relative changes in protein phosphorylation based on the densitometric immunoblotting analysis (Supplementary Fig. 1h) of cell lysates of adipocytes from α4KD and Control for 5 min using 100 nM insulin. Data are presented as mean ± SEM (One-way ANOVA post hoc Bonferroni test; three clones were used in three independent experiments, p-IR: ***$p = 0.0007$, p-Akt (S473): **$p = 0.001$, p-S6 (S235/S236): ***$p = 0.0001$). Source data are provided as a Source Data file.

cells, α4 KO primary brown preadipocytes showed a 55% decrease in insulin-stimulated IR Tyr phosphorylation and a parallel significant decrease in Akt phosphorylation (Fig. 2d). As expected, based on its known interaction[28], α4 deletion also caused a 60% decrease in the PP2Ac protein level (Fig. 2e). Interestingly, YBX1 phosphorylation in both the basal and the insulin-stimulated state was enhanced in the α4KO preadipocytes while total YBX1 protein levels remained unchanged (Fig. 2d). We hypothesized that α4 interacting with YBX1 may regulate *Ptp1b* transcription. Consistent with a previous report that YBX1 is a positive regulator of PTP1B expression[36], α4KO preadipocytes showed a two-fold enhancement of PTP1B expression in both the basal and insulin-stimulated states (Fig. 2d). To further confirm the effect of α4 on PTP1B expression, we employed a reporter system using a *Ptp1b* promoter-driven luciferase vector as described by Fukada and Tonks[36]. While YBX1 dramatically enhanced the promoter activity of *Ptp1b*, overexpression of α4 significantly blunted the transactivation ability of YBX1 on *Ptp1b* promoter in HEK293 cells (Fig. 2e). These results indicate that α4 can play a major role in the attenuation of insulin signaling by influencing the interaction between Ser/Thr phosphatases and YBX1, eventually modulating PTP1B expression, leading to a feedback regulation of IR Tyr phosphorylation (Fig. 2f).

## Acute adipocyte-specific α4 knockout significantly alters lipid composition in WAT and BAT

To confirm the role of α4 in insulin signaling and adipogenesis in vivo, we established a mouse model (Ai-α4KO) in which α4 was specifically deleted in adipocytes in an inducible manner by crossing α4$^{flox/flox}$ mice and Adipoq-CreER$^{T2}$ mice[37]. Ai-α4KO mice displayed 85 and 82% decreases in α4 at the protein level in BAT and iWAT depots, respectively, as early as 3 days after the last tamoxifen injection (Supplementary Fig. 3a, b). At this time point, although Ai-α4KO mice did not show any obvious change in weight of adipose tissue in different depots or adipocyte cell diameter as compared with Controls (Fig. 3a–c), PTP1B expression was significantly upregulated in BAT and iWAT extracts from Ai-α4KO mice compared with Controls (Fig. 3d, e and Supplementary Fig. 3c). Consistent with the in vitro experiments, Ai-α4KO mice showed impaired insulin signaling in adipose tissues, showing 47–70% decreases in IR Tyr phosphorylation, Akt phosphorylation and PP2Ac expression in both BAT and iWAT (Fig. 3d, e and Supplementary Fig. 3d, e). The impaired Tyr phosphorylation of IR in Ai-α4KO adipocytes occurred at the very early stage, when tissue inflammation and adipocyte apoptosis were not detected, suggesting that the impaired IR signaling was not due to adipose tissue damage (Fig. 3f and Supplementary Fig. 3f). At day 6, Ai-α4KO mice showed a mild but significant decrease in adipocyte cell diameter in iWAT, as well as eWAT, and irregularity in adipocyte size in BAT, indicative of

alteration of the content and composition of structural lipids in WAT and BAT due to the continued impairment of insulin signaling in Ai-α4KO mice (Fig. 3g, h and Supplementary Fig. 3g). Moreover, the impairment of insulin signaling in Ai-α4KO was associated with a significant increase in rates of basal lipolysis as measured using explants of iWAT and eWAT, with no significant change of maximal isoproterenol-stimulated lipolysis (Fig. 3i and Supplementary Fig. 3h).

To assess the content and composition of structural lipids in iWAT, BAT, and serum, we employed global lipidomics using liquid chromatography coupled with tandem mass spectrometry (LC-MS/MS), and the results are shown in the heatmaps in Fig. 3j and Supplementary Fig. 3i–m. The results revealed significant increases in the levels of cholesterol esters (CE), sphingomyelin (SM), ceramide (CER), and especially hexosylceramide (HCER) in both BAT and iWAT of Ai-α4KO as compared with Controls (Fig. 3k and Supplementary Fig. 3n). Among the ceramide species, Ai-α4KO mice showed a notable increase in the level of C16:0 (Fig. 3l and Supplementary Fig. 3o), which has been shown to act as a mediator of insulin resistance in both the BAT and iWAT[11,38,39]. iWAT of Ai-α4KO also showed a significant increase in the overall concentration of the five classes of phospholipids, phosphatidylcholine (PC), phosphatidylethanolamine (PE), phosphatidylinositol (PI), lysophosphatidylcholine (LPC), and lysophosphatidylethanolamine (LPE), while BAT showed no changes of these lipids (Fig. 3k and Supplementary Fig. 3n). In addition, triacylglycerols (TAGs), the most abundant lipid class and the primary source of stored energy in WAT and BAT, was significantly decreased in both iWAT and BAT of Ai-α4KO mice (Fig. 3k and Supplementary Fig. 3n). The profiles of fatty acids (FAs), including saturated fatty acids (SFAs), monounsaturated fatty acids (MUFAs), and omega-3 (ω−3) polyunsaturated fatty acids (PUFAs), such as eicosapentaenoic acid (EPA; C20:5) and docosahexaenoic acid (DHA; C22:6), which have been suggested to counteract insulin resistance by mediating anti-inflammatory effects[40], showed significantly lower levels in both the iWAT and BAT of Ai-α4KO mice (Fig. 3m and Supplementary Fig. 3p).

## Acute α4 deletion in adipocytes results in impaired mitochondrial gene expression and decreased adaptive thermogenesis

To investigate the full impact of α4 deletion in adipocytes, total mRNAs in iWAT and BAT from Ai-α4KO mice one week after recombination were subjected to RNA-seq analysis. Principal component analysis demonstrated a global change in gene expression in both iWAT and BAT from Ai-α4KO mice (Fig. 4a). Of the more than 14,000 transcripts detected, 1897 genes were regulated by both Ai-α4KO iWAT and BAT. Of these, 1293 and 604 genes were upregulated and downregulated by at least two-fold in both iWAT and BAT, respectively (FDR < 0.01, for both groups), indicating the broad impact of α4 on

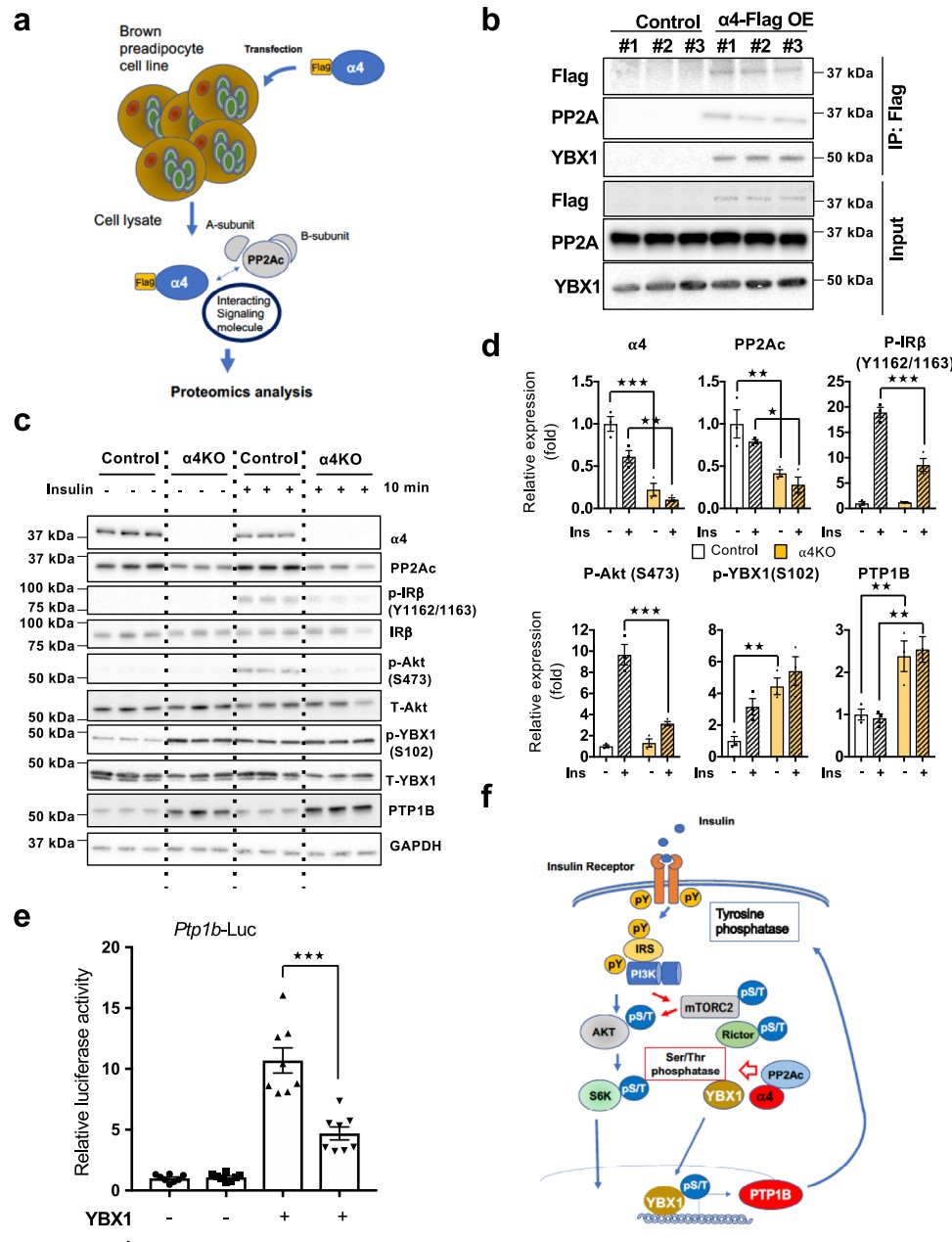

**Fig. 2 | α4 regulates IR Tyr phosphorylation via YBX1 and PTP1B. a** Schematic diagram showing the regulation of insulin signaling in brown preadipocytes via α4. Specifically, α4 binds to the catalytic subunit of protein phosphatase 2A (PP2Ac) and is known to regulate Ser/Thr phosphorylation. For the mass spectroscopic proteomic analysis. we compared the Control ($n = 1$) and α4 overexpressing ($n = 2$, biological replicates) samples. **b** α4-complex containing YBX1. Flag-tagged α4 immunoprecipitation showed the interaction between YBX1 with α4 as well as that between PP2Ac and α4 in brown preadipocytes ($n = 3$ biologically independent cell clones/group). **c** Western blotting of phosphorylation of insulin signaling molecules in Control and α4KO brown preadipocytes following indicated concentrations of insulin stimulation for 10 min. **d** Effect of α4 on insulin signaling-based densitometric immunoblotting analysis with antibodies to α4 and phosphorylated IR, Akt, YBX1, as well as PTP1B in lysates of α4 Knockout brown preadipocytes (α4 KO) before and 10 min after insulin stimulation (see **c**). Data are presented as mean ± SEM (one-way ANOVA post hoc Bonferroni test, $n = 3$, each cell was

generated from biologically independent animals from three independent experiments, Statistical significance is shown as $p < 0.05$ (*), $p < 0.01$ (**), and $p < 0.001$ (***)). **e** Luciferase reporter assay for PTP1B transcription was performed as described by Fukada and Tong[45]. Cells were transfected with the expression vectors of α4 and/or YBX1 in the presence of *Ptp1b*-Luc reporter or mock plasmid DNA in HEK293cells. Data are presented as mean ± SEM (One-way ANOVA post hoc Bonferroni test, $n = 8$ technical replicates per group, ***$p < 0.0001$). **f** Model of the regulation of α4 in insulin-activated IR phosphorylation. Insulin-bound IR phosphorylates itself and IRS1/2, and activates the PI3K-AKT and S6K pathways. α4 stabilizes PP2A and dephosphorylates mTORC2 with Rictor. α4 also binds to Y-box protein 1 (YBX1), which functions as a transcription factor for Tyr phosphatase PTP1B. α4 blocks YBX1 nuclear translocation and inhibits PTP1B expression, thus in feedback regulation, promoting the enhanced signaling through the insulin receptor. Source data are provided as a Source Data file.

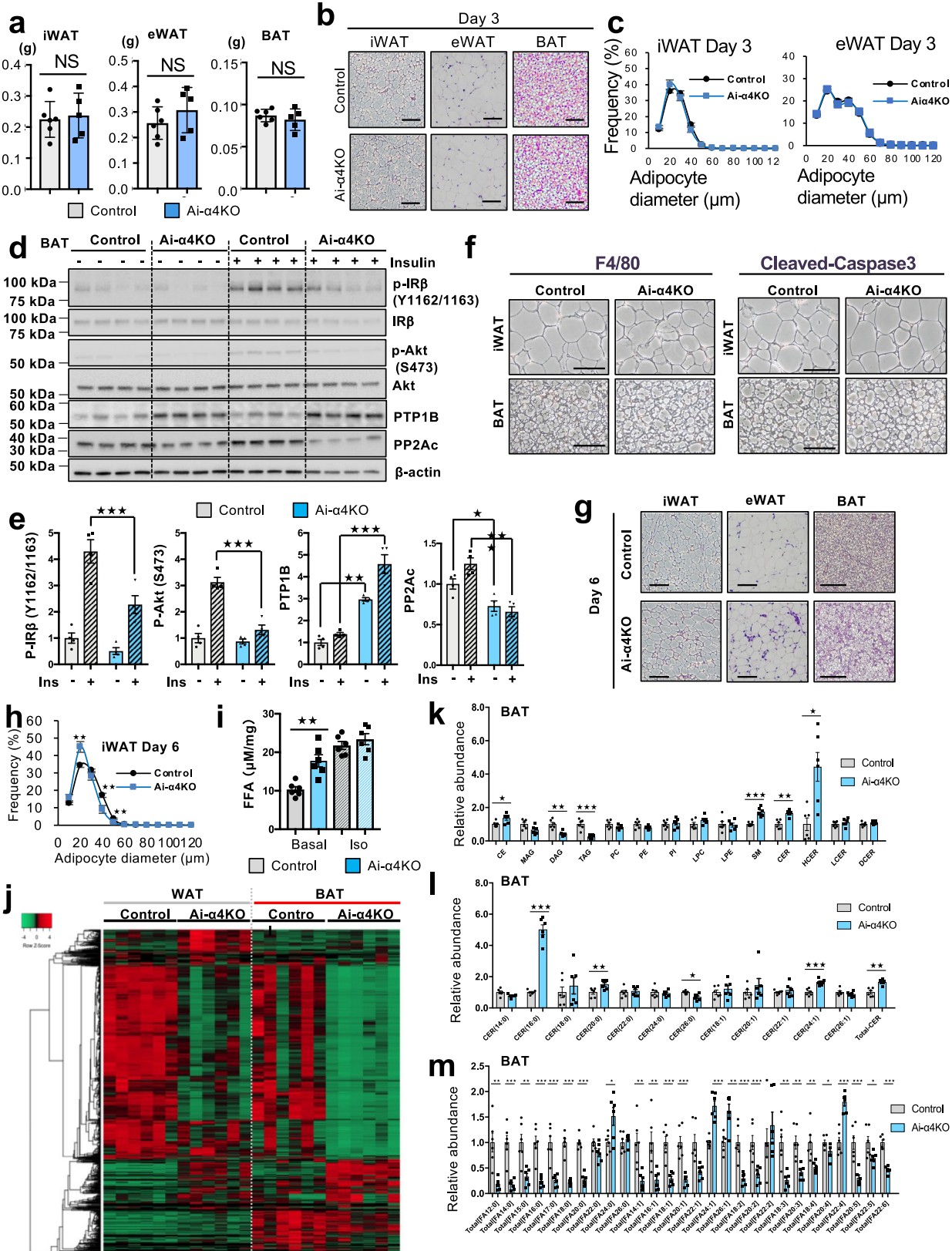

adipocyte biology (Fig. 4b). Further, among these transcripts, 410 and 450 genes were specifically upregulated and downregulated in Ai-α4KO iWAT only, while 1180 and 391 genes were upregulated and downregulated in Ai-α4KO BAT only (Fig. 4b). These transcripts included genes associated with metabolism, insulin signaling and non-coding genes and miRNAs. Differentially expressed genes in Ai-α4KO

compared with Control in each depot are shown in heatmap form in Fig. 4c, d and in a volcano plot in Supplementary Fig. 4a.

The top-ranked commonly downregulated mRNAs in both iWAT and BAT from Ai-α4KO were the transporter for glucose (*Slc2a5*) and Acyl-Coenzyme A Synthetase (*Acsm3*) (Fig. 4c), while lipase family member N (*Lipn*), a gene involved in lipoprotein metabolism, was

**Fig. 3 | Role of α4 in the regulation of lipid metabolism in mature adipocytes.** **a** Tissue weights (g) of iWAT, eWAT, and BAT from Control ($n = 6$) and Ai-α4KO ($n = 5$) male mice on day 3 after treatment with tamoxifen (Two-tailed Student's $t$-test). **b** HE-stained sections of iWAT, eWAT, and BAT from Control and Ai-α4KO mice on day 3. Scale bars = 100 μm. **c** Diameter distribution of isolated iWAT and eWAT adipocytes in Control and and Ai-α4KO at day 3. Data are mean ± SEM (two-tailed Student $t$-test, iWAT; Control ($n = 8$), Ai-α4KO ($n = 7$), eWAT; ($n = 9$/group). **d** Immunoblotting results showing the insulin signaling of phosphorylated IR, Akt, and PTP1B in interscapular BAT of 12-wk-old fasted Control and Ai-α4KO mice 10 min after i.v. insulin stimulation (5 IU per mouse) or control saline ($n = 4$). **e** Densitometric comparison of the phosphorylation signals shown as fold increases in relation to the control without insulin stimulation (1.0). Data are presented as mean ± SEM (one-way ANOVA post hoc Bonferroni test: $n = 4$, *$p < 0.05$, **$p < 0.01$ and ***$p < 0.001$, biologically independent animals per group from one experiment). **f** Staining of iWAT and BAT sections from Control and Ai-α4KO mice on day 3 for F4/80 or cleaved caspase-3. The experiments were repeated independently four times. Scale bars = 50 μm. **g** HE-stained sections of iWAT, eWAT, and BAT from Control and Ai-α4KO mice on days 6. Scale bars = 100 μm. **h** Diameter distribution of isolated iWAT from Control and Ai-α4KO mice on day 6. Data are presented as mean ± SEM (Two-tailed Student's $t$-test: $n = 8$/group). **i** Lipolysis assessed by FFA release from iWAT of Control and Ai-α4KO mice on day 6. Samples were incubated ex vivo in the presence or absence of 10 mM isoproterenol, and FFA release into the medium was quantified. (Two-tailed Student's $t$-test, **$p = 0.002$: $n = 6$/group). **j** Heatmap showing Z-scores of lipid species in iWAT and BAT from Control and Ai-α4KO mice ($n = 5$/group) at 1 wk. Green or red represents a decrease and increase, respectively. Comparison of lipid classes (**k**), fatty acids with different chain lengths (**l**) and Ceramide species with the indicated chain lengths **m** in BAT from Ai-α4KO mice with respect to the 1.0 level corresponding to the Control ($n = 5$/group). Data are presented as mean ± SEM (Two-tailed Student's $t$-test). Statistical significance is shown as $p < 0.05$ (*), $p < 0.01$ (**), and $p < 0.001$ (***). Source data are provided as a Source Data file.

among the most upregulated in both iWAT and BAT (Fig. 4c, d). Gene set pathway analysis revealed that the most downregulated pathways in Ai-α4KO mice were related to the metabolic pathways, oxidative phosphorylation and the citrate cycle (TCA cycle) (Fig. 4e and Supplementary Fig. 4b, c). Further, both iWAT and BAT from Ai-α4KO mice showed a decrease in the expression levels of multiple genes that are associated with mitochondrial oxidative phosphorylation (OXPHOS), such as the 21 components of the NADH dephosphorylase 1 subunit complex (*Ndufab1, Ndufa1, Ndufa4, Ndufa6, Ndufa9, Ndufa10, Ndufa11, Ndufa12, Ndufb2, Ndufb5, Ndufb6, Ndufb8, Ndufb9, Ndufb10, Ndufb11, Ndufs1, Ndufs2, Ndufs3, Ndufs5, Ndufs7,* and *Ndufs8*), ATPase H+ transforming subunits (*Atp6v1g1, Atp6v1b2, Atp6v1f, Atp6v0b, Atp6v0a1, Atp6v0a2,* and *Atp6v0a4*), ATP synthases (*ATP5a1, ATP5,* and *ATP5d*), cytochrome C oxidase subunits (*Cox5b, Cox6a1, Cox6c, Cox7a1, Cox7a2, Cox7b, Cox8a,* and *Cox8b*), succinate dehydrogenase complexes (*Sdha, Sdhb, Sdhc,* and *Sdhd*), ubiquinol cytochrome C reductase binding proteins (*Uqcrfs1, Uqcrc1, Uqcrs2, Uqcrq, Uqcr11,* and *Uqcr1*), and the enzymes of the TCA cycle genes (*Idh1, Idh2, Idh3a, Idh3b, Mdh1,* and *Mdh2*) (Fig. 4f and Supplementary Fig. 4b). These results indicate the changes occurred not only in BAT but also in iWAT containing beige adipocytes from Ai-α4KO mice. A heatmap of the top 50 regulated genes (Fig. 4d) showed that the expression of *UCP1*, which transports protons directly from the mitochondrial intermembrane space into the cell matrix, was significantly decreased in iWAT from Ai-α4KO mice. UCP1 protein levels as detected by immunostaining was decreased in both iWAT and BAT from Ai-α4KO mice compared with Controls (Fig. 4g). On electron microscopy, the mitochondria in both white and brown adipocytes of Ai-α4KO mice displayed abnormal morphology with irregular and reduced cristae organization in comparison with the Control mitochondria indicating the impairment of the mitochondria in brown adipocytes and also in iWAT containing beige adipocytes (Fig. 4h). The average size of an individual mitochondrion was decreased by 22% in brown adipocytes from Ai-α4KO (Fig. 4i).

Consistent with the changes in mitochondrial morphology and mitochondrial gene expression, α4 deletion affects cell respiration and thermogenic adaptation in vivo. Oxygen consumption rates (VO2) were decreased by 13–14% during the dark cycle and 21–22% during the light cycle in Ai-α4KO mice compared with Controls on standard chow diet (Fig. 4j). Even though both the Control and Ai-α4KO mice had similar basal body temperature (~37 °C), when placed at 4 °C, Ai-α4KO mice were unable to maintain their core (rectal) and BAT (interscapular) temperatures (Fig. 4k and Supplementary Fig. 4d). This was confirmed by thermal imaging of the mice which demonstrated lower body temperature compared with Controls over the interscapular region on day 10 after gene targeting (Fig. 4l). Together these results showed that α4 is indispensable for adipocyte mitochondrial function and thermogenesis in mice.

## Acute α4 knockout showed adipocyte apoptosis and adipose tissue inflammation

Further analysis of the gene transcription profiles revealed that the most upregulated genes in both BAT and iWAT from Ai-α4KO compared to controls were those of cytokines, cytokine receptors, NF-κB signaling and TNF signaling pathways (Fig. 5a, b and Supplementary Fig. 5a–d). Thus, BAT and iWAT of Ai-α4KO mice showed increased levels of proinflammatory cytokines (*Tnf-α* and *Il-6*) and chemokines, such as *Ccl2/Mcp-1* (C-C motif chemokine ligand 2/macrophage chemoattractant protein 1), *Ccl5*, and the C-X-C motif chemokine ligand 10, *Cxcl10* - all of which are involved in adipose macrophage infiltration and insulin resistance pathogenesis (Fig. 5c and Supplementary Fig. 5e–g). Consistent with the increased expression of *CD11c* and *F4/80* (Fig. 5d), both white and brown fat from Ai-α4KO mice showed massive macrophage infiltration into adipose tissues (Fig. 5e). Flow cytometry analysis also revealed that the ratio of M1 (CD11c⁺) to M2 (CD206⁺) macrophages in iWAT and BAT from Ai-α4KO mice were significantly higher than those in iWAT and BAT from the Control mice (Fig. 5f and Supplementary Fig. 5h–i). This macrophage infiltration was associated with increased levels of cleaved caspase-3 and increased TUNEL staining in both iWAT and BAT of the Ai-α4KO mice, indicating increased adipocyte apoptosis on days-6 and -9 after tamoxifen administration (Fig. 5g, h and Supplementary Fig. 5j–l). Moreover, there was upregulation of genes related to the NF-κB signaling pathway, such as *Tlr4, Myd88, Nfkb1,* and *Nfkb2*, indicative of macrophage *Nlrp3* inflammasome activation, which in turn led to the induction of *caspase-1* and promotion of the expression of *Il-1β* and *Il-18* in Ai-α4KO mice (Fig. 5i).

## Adipose tissue recovery from the acute adipose tissue damage in Ai-α4KO mice

To study the long-term consequences of increased adipocyte apoptosis in Ai-α4KO mice, we analyzed WAT and BAT on days 9 and 17 after tamoxifen administration. While Ai-α4KO mice showed a modest decrease (24–31%) in iWAT, eWAT, and BAT depots on day 9, these decreases became greater by day 17 (37–63%) (Fig. 6a and b). Accompanied by a progressive decrease in fat mass, analysis of gene expression in iWAT on day 17 showed 63–99% decreases in the expression levels of *adiponectin, Leptin, PPARγ, AP2, Glut4, FAS, ATGL, CEBPα, Elovl3, Tfam, Adrb3, PRDM16,* and *UCP1* mRNAs in Ai-α4KO mice (Fig. 6c). Expression of these adipocyte markers was also decreased in Ai-α4KO BAT by 64–77%, demonstrating continued impairment of the transcriptional programming required for the maintenance of fully differentiated adipocytes (Fig. 6d). Histologically, Ai-α4KO mice on day 9 displayed an increased number of crown-like structures, indicating macrophages surrounding dying or dead adipocytes in iWAT and BAT (arrows in Fig. 6e, f). By day 17, Ai-α4KO mice showed extensive loss of adipocytes in all the depots as well as a

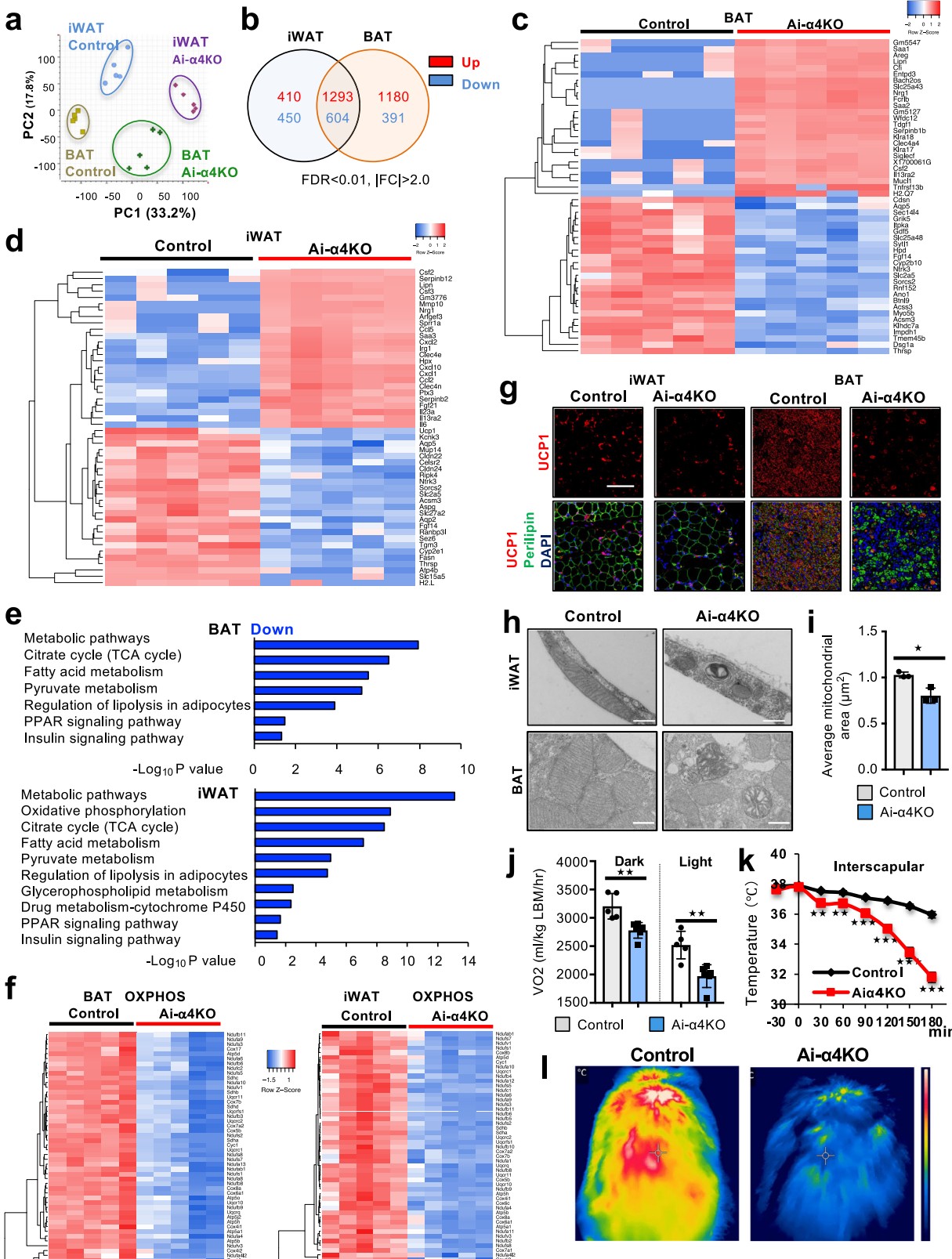

dramatic increase in stromovascular cell numbers in iWAT and BAT. However, this process was reversible, such that by day 30, the widespread adipocyte apoptosis disappeared in Ai-α4KO mice (Fig. 6e, f), and by day 74, Ai-α4KO WAT fully recovered, showing a normal morphology, while Ai-α4KO BAT contained multilocular fat cells, which are typical of BAT as well as large unilocular fat cells that resembled WAT

during the recovery period (Fig. 6e, f). Both the multilocular and unilocular cells in Ai-α4KO BAT were positive for UCP1 protein, whereas Ai-α4KO WAT showed a homogenous cell population with no UCP1 positive adipocytes (Fig. 6g and Supplementary Fig. 6a).

To assess the rate of adipocyte regeneration, we performed lineage tracing by crossing mice of Ai-α4KO or adiponectin-CreER[T2]

**Fig. 4 | Role of α4 in the regulation of gene expression and mitochondrial biogenesis in mature adipocytes. a** Principal component analysis plots of the transcriptome profiles of Control iWAT (Blue), Ai-α4KO iWAT (purple), Control BAT (Gold), and Ai-α4KO BAT (Green). **b** Venn diagram showing the numbers of significantly regulated genes in Ai-α4KO iWAT and BAT compared with the Control iWAT and BAT (FDR < 0.05, |FC| > 2.0). Heatmap of top 50 (up and down: top 25 each) differentially regulated genes between Control and Ai-α4KO in BAT (**c**) and iWAT (**d**). **e** Top directionally downregulated KEGG pathways between Control and Ai-α4KO in BAT (upper) and iWAT (bottom). **f** Genes involved in oxidative phosphorylation (OXPHOS) in BAT (left) and iWAT (right) listed in the heatmap. The color intensities indicate the Z-score of each gene. **g** UCP1 expression in iWAT and BAT from Control and Ai-α4KO mice at 1 wk ($n = 3$ biologically independent animals per group). Scale bars = 100 μm. **h** Representative electron microscopic images of mitochondria in white and brown adipocytes from Control and Ai-α4KO mice 1 wk after tamoxifen administration. Scale bars = 500 nm. **i** Quantification of the average mitochondrial size from Control and Ai-α4KO brown adipocytes 1 wk after tamoxifen administration. Data are mean ± SEM (Two-tailed Student's $t$-test: *$p = 0.01$; $n = 3$). **j** Oxygen consumption (VO2) corresponding to Control ($n = 5$) and Ai-α4KO ($n = 6$) mice housed in metabolic cages 1 wk after tamoxifen administration. The dark phase represents the 12-h period of a day during which the lights were turned off. Data are mean ± SEM (Two-tailed Student's $t$-test, Dark: **$p = 0.003$, Light: **$p = 0.002$). **k** Interscapular temperature of Control ($n = 6$) and Ai-α4KO ($n = 5$) mice on day 10 during a 3-h exposure to an environment at 4 °C. Data are mean ± SEM (Two-tailed Student's $t$-test: *$p < 0.05$; **$p < 0.01$; ***$p < 0.001$). **l** Thermal images showing the comparison of the surface temperature over interscapular BAT from Control and Ai-α4KO mice after 2-h exposure to a temperature of 4 °C on days 10 and 90. Source data are provided as a Source Data file.

transgenic, with mice of mTmG reporter tandemly aligned cDNAs corresponding to membrane-targeted tomato-fluorescent protein (mTFP) and membrane-targeted green fluorescent protein (mGFP)[41] (Supplementary Fig. 6b). In the Control adiponectin-creER^T2:mTmG mice, within 1 wk after tamoxifen administration, more than 95% of the adipocytes exhibited green fluorescent staining via tamoxifen-induced gene recombination (Fig. 6h–j). 12 weeks after tamoxifen administration, Control-mTmG mice carrying the Adipoq-Cre-ER^T2 transgene continued to display green-labeled adipocytes in both iWAT, eWAT, and BAT, indicating slow turnover rates of adipocytes in animals under normal conditions. However, expression of GFP gradually disappeared by Week 4, and in parallel, red fluorescent protein marked adipocytes with both unilocular and multilocular lipid droplets gradually became the dominant type of adipocytes in iWAT, eWAT, and BAT from the same mice (Fig. 6h–j). This replacement by new adipocytes became even more prominent by Weeks 8 and 12 after tamoxifen administration (Fig. 6h–j), indicating continuous adipocyte regeneration during the recovery phase of adipose tissues in Ai-α4KO WAT and BAT. Sensitivity to cold exposure corresponded with the recovery of BAT (Fig. 6k). With this, there was also functional changes in temperature regulation. Thus, on day 10, Ai-α4KO mice showed smaller BAT mass (Fig. 6a) and lowered *UCP1* expression (Fig. 4g) and high sensitivity to cold exposure (4 °C), with a marked decrease in body temperature to ~30 °C after 3 h (Figs. 4k, l and 6l). By day 90, Ai-α4KO mice recovered and displayed cold resistance to a nearly normal level, observing the gradual but significant recovery of the functional BAT (Fig. 6k, l and Supplementary Fig. 6c).

## α4 is essential for adipose tissue development and metabolism in high-fat diet-fed mice

To investigate whether α4 is important for the normal development and maintenance of adipose tissues, we investigated the phenotype of constitutive α4 deletion by using Adipoq-Cre mice (Aα4KO) (Fig. 7a). On chow diet, body weights of Aα4KO mice were similar to those of mice in the Control group at up to 10 weeks of age, but by 16 weeks, the KO mice were about 12% heavier than the Controls (Fig. 7a). Based on CT imaging, Aα4KO mice exhibited less than 1% of the amount of visceral and subcutaneous fat compared with Controls (Fig. 7b and Supplementary Fig. 7a), and this was confirmed by dissection which revealed an almost complete loss of eWAT and > 90% decreases in iWAT and BAT, indicating that α4 is necessary for the development of adipose tissues.

In response to the severe loss of WAT and BAT, Aα4KO mice showed enlarged livers with an average two-fold increase in liver weight, presumably secondary to uptake and accumulation of circulating lipid contents (Fig. 7c). Consequently, Aα4KO mice developed hepatomegaly associated with increased intrahepatic triglyceride accumulation and hepatocyte balloon degeneration, the former contributing to the increased body weight observed at 11 weeks of age (Fig. 7c and Supplementary Fig. 7b–d). The livers of

KO mice also showed 2- to 3-fold increases in the expression of genes involved in de novo lipogenesis (*Srebp1c*, *Scd1*, and *Fasn*), as well as chemokines, such as CCL2, indicating liver inflammation (Fig. 7d). Consistent with the loss of BAT mass, the mice also failed to maintain their body temperatures when placed in an environment at a temperature of 4 °C for 3 h, dropping their rectal and interscapular temperature by 15.2 and 16.7 °C, respectively (Supplementary Fig. 7e, f). Furthermore, the Aα4KO mice showed a significant decrease in oxygen consumption rates (VO2) (Supplementary Fig. 7g). They also developed a severe metabolic syndrome, with an average fasting blood glucose level of ~140 mg/dl and a fed glucose level >230 mg/dl (Fig. 7e), associated with an 8.9-fold increase in serum insulin levels (Fig. 7f). In addition, Aα4KO mice displayed severe glucose intolerance during an intraperitoneal glucose tolerance test and marked resistance to exogenous insulin upon insulin tolerance testing (Supplementary Fig. 7h, i). Consistent with the marked lipodystrophy, Aα4KO mice showed >89% decreases in circulating leptin levels (Fig. 7f). Importantly, leptin administration via subcutaneous infusion pumps prevented the development of hyperglycemia in Aα4KO mice (Fig. 7g), indicating that leptin is able to prevent the metabolic abnormality associated with lipodystrophy, similar to other lipodystrophy mouse models[4].

To elucidate the effect of Aα4KO on the development of obesity and metabolism of the Aα4KO mice, we challenged Control and Aα4KO mice with a 60% high-fat diet (HFD) for 4 months. On this diet, control mice showed a 50 % increase in body weight compared to mice on normal chow, whereas Aα4KO mice showed only a 10 % increase in weight gain even with HFD (Fig. 7h). Accordingly, Aα4KO mice showed ~99 % reductions in iWAT and eWAT respectively, after the HFD, indicating that Aα4KO mice were resistant to diet-induced obesity owing to the inhibition of adipocyte expansion and regeneration (Supplementary Fig. 7j). The HFD-fed Aα4KO mice showed a 78% decrease in BAT mass compared with Control mice on HFD (Supplementary Fig. 7j). Despite the resistance to obesity, the HFD-fed Aα4KO mice showed severe hyperglycemia even after a 6-h fast (Fig. 7i), with high blood glucose levels (231 ± 20 mg/dl) as compared to those in HFD-fed Control mice (160 ± 9 mg/dl). Additionally, the HFD-fed Aα4KO mice showed 2.6-fold higher serum insulin levels than the HFD-fed Control mice (Supplementary Fig. 7k). This was associated with elevated glucose levels during an intraperitoneal glucose tolerance test and a marked impairment of response based on the results of an intraperitoneal insulin tolerance test in HFD-fed Aα4KO mice (Supplementary Fig. 7j, k). Accompanying this systemic insulin resistance, HFD-fed Aα4KO mice showed even more markedly increased islet mass and β-cell proliferation compared to controls on HFD, as well as increased Ki67 staining, indicating an even higher level of β-cell replication (Fig. 7l and Supplementary Fig. 7l). Going along with the loss of adipose tissues in HFD-fed Aα4KO, hepatomegaly was profound, with a 1.7-fold increase in liver weight and a 1.9-fold increase in liver triglyceride accumulation in the HFD-fed Aα4KO mice compared

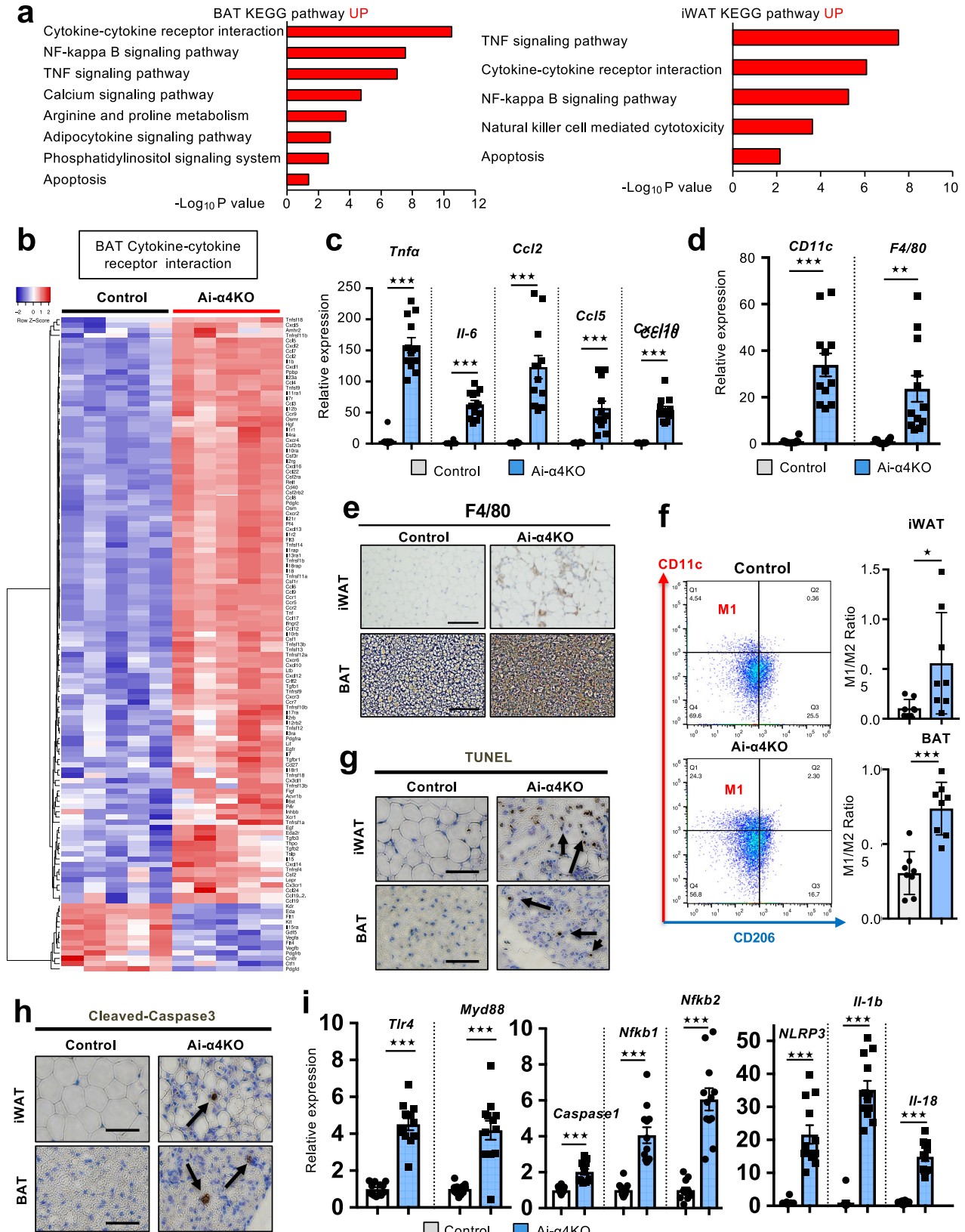

with those in HFD-fed Control mice (Fig. 7m and Supplementary Fig. 7m). Histological examination of the liver revealed marked micro- and macro-vesicular steatosis and balloon degeneration in Aα4KO mice (Supplementary Fig. 7n). Ionizing calcium-binding adapter molecule 1 (Iba-1) immunostaining and Azan staining showed that the HFD-challenged Aα4KO mice developed liver inflammation and

fibrosis, with elevated *IL-1β*, *CCL2*, and *TGF-β1* mRNA levels (Fig. 7n, o, and Supplementary Fig. 7o). Overall, despite being protected from apparent obesity, feeding Aα4KO mice with a HFD accelerated diabetes, progressive NAFLD, and pancreatic islet hyperplasia. Taken together, these lipodystrophic mutant mice developed liver injury secondary to lipotoxicity, and HFD feeding exacerbated the liver

**Fig. 5 | Adipose tissue inflammation in Ai-α4KO mice. a** Top upregulated KEGG pathways between Control and Ai-α4KO in BAT (left) and iWAT (right). **b** Heatmap of genes involved in the cytokine-cytokine receptor interaction, with the color intensities indicating the Z-score of each sample. Expression levels of inflammatory cytokines, chemokines (**c**), and macrophage (**d**) markers in BAT from Control and Ai-α4KO mice on day 9 (n = 12/group). (Two-tailed Student's t-test, **p < 0.01; ***p < 0.001). **e** Staining of iWAT and BAT sections from Control and Ai-α4KO mice on day 9 for F4/80. Scale bars = 50 μm. **f** Flow cytometric analysis results showing infiltrated immunocytes with CD11c and CD206 on day 9. Representative data corresponding to iWAT are indicated as the FACS profiles (left). The ratio of M1 (CD11c$^+$) to M2 (CD11c$^+$) macrophages infiltrated in iWAT and BAT are shown in the right graph (n = 8/group). (Two-tailed Student's t-test, *p = 0.03, ***p < 0.0001). **g** TUNEL staining results corresponding to iWAT and BAT sections from Control and Ai-α4KO mice on day 9. Scale bars = 50 μm. The arrow shows TUNEL-positive cells. **h** Immune-stained iWAT and BAT sections from Control and Ai-α4KO mice for the identification of cleaved caspase-3 on day 9. Scale bars = 50 μm. The arrow shows cleaved caspase-3 cells. **i** Expression levels of genes related to NF-κB signaling pathways and inflammasome components measured via real-time qPCR using BAT from Control and Ai-α4KO mice on day 9 (n = 12/group). Data are presented as mean ± SEM (Two-tailed Student's t-test, ***p < 0.0001). Source data are provided as a Source Data file.

disease, eventually causing altered whole-body metabolism and overt disease pathogenesis.

## Discussion

Insulin is a major regulator of glucose uptake and lipid metabolism in mature adipocytes and plays an important role in the development and maintenance of both WAT and BAT. In the present study, we have discovered the phosphatase-associated protein α4 as a critical regulator of IR signaling and adipose tissue homeostasis.

Although its primary role in cells has been viewed as a regulator of the Ser/Thr phosphatases PP2A, α4 depletion in adipocytes resulted in a prominent decrease in both insulin-stimulated IR and Akt activation. The impairment of insulin signaling in α4KO adipose tissues occurs as early as 3 days after tamoxifen treatment, when no detectable evidence of inflammation and apoptosis in adipose tissues was found, suggesting a direct regulatory loop between α4 and insulin signaling in adipose tissues. Our mechanistic studies uncovered that the α4-PP2A complex associates with the transcription factor YBX1, inhibits the expression of the Tyr phosphatase PTP1B, and subsequently potentiates IR Tyr phosphorylation in adipocytes. While several tyrosine phosphatases have been shown to act on the insulin receptor, the cytoplasmic protein Tyr phosphatase PTP1B is the primary negative regulator of Tyr phosphorylation of the activated IR, IGF1R, and IRS proteins[25]. Thus, loss of PTP1B in mice results in increased IR Tyr phosphorylation and enhanced insulin sensitivity in the muscle and liver, as well as resistance to HFD-induced obesity[42,43]. Additionally, the HFD induces the expression of *Ptp1b* mRNA in the adipose tissues, liver, and skeletal muscles of mice further contributing to insulin resistance[44]. TNFα has also been shown to upregulate the expression of *Ptp1b* mRNA in cell lines and mice, contributing to induce insulin resistance[44].

At a transcriptional level, the *Ptp1b* promoter has two regulatory elements: an enhancer, which can be recognized by YBX1, and a p210 Bcr-Abl PRS motif, which is recognized by transcription factor-s in the Sp family[45]. Overexpression of YBX1 results in increased *Ptp1b* mRNA levels[36]. In cancer cell lines, the transcription activity of YBX1 is controlled by the phosphorylation status of Ser residues 102 or 165[46]. The activity of PTP1B is also regulated by the phosphorylation of Ser[47] and Tyr[48,49] residues. At the cellular level, PTP1B acts preferentially on receptor-protein Tyr kinases (RTKs) that have undergone endocytosis, indicating that it plays a role in the downregulation of the growth factor signal rather than controlling the basal phosphorylation status of the receptor[50].

We investigated the impact of the regulation of IR signaling via α4 in adipose tissues in vivo using two kinds of gene targeting strategies. The first was the inducible deletion of α4 in mature adipose tissues using tamoxifen-activated adiponectin-Cre ER$^{T2}$ mice (Ai-α4KO). Ai-α4KO resulted in a rapid alteration of lipid profiles with elevated levels of adipocyte ceramides, especially those of C16:0 and C18:0 species. This was accompanied by unrestrained lipolysis and a more gradual development of inflammation and adipocyte apoptosis, as indicated by the observed increase in intensity of TUNEL and cleaved caspase-3 staining, decreased mitochondrial function, and a gradual decrease in

adipose tissue mass. Adipose tissues from Ai-α4KO mice showed macrophage accumulation and increased proinflammatory gene expression. Further, adipose tissues from Ai-α4KO mice showed a significant decrease in the level of omega-3 fatty acids (ω3-FAs), including EPA and DHA, as well as decreased NLRP3 and NLRP1β-dependent caspase-1 activation and IL-1β secretion. Recent studies revealed that ω3-FAs suppress inflammation via the inhibition of inflammasome activation[51]. Several lines of evidence support the effect of ceramides as the primary membrane lipids involved in insulin resistance alongside diacylglycerols[8]. At the molecular level, ceramides can impair Akt activation and GLUT4 translocation to the cell membrane[52]. Thus, an increase in inflammation, as well as an increase in the levels of ceramides, could further exacerbate insulin resistance in α4-deficient adipose tissues.

Ai-α4KO mice presented marked cold intolerance, which was associated with the impairment of mitochondrial function in the WAT and BAT of the mice as early as day 7. In this model, however, there is a reversal of cold intolerance by day 90 after tamoxifen treatment associated with the regeneration of both WAT and BAT. The regenerated BAT, however, shows altered morphology, containing a mixture of cells with multilocular and unilocular fat droplets, as seen in brown or beige adipocytes. This heterogeneity might represent regeneration of BAT from heterogeneous populations of precursors, including a Myf-5-positive lineage like classical brown adipocytes[53]. Several other mouse models with inducible lipodystrophy have shown similar adipose tissue regeneration capacity. For example, when adipose tissues are acutely ablated in a Fat-ATTAC and the fat-specific inducible IR/IGF1R DKO mouse models, there is apoptosis in adipose tissues accompanied by the activation of caspase-8[4,54]. In both models, following the initial rapid loss of fat, the induction of preadipocyte differentiation occurs, producing new populations of functional brown and white adipocytes to restore fat tissues and resolve the metabolic syndrome, demonstrating the possibility of adipose tissue regeneration. This homeostatic capacity of adipose tissues may be the result of an as of yet undefined adipotrophic factor(s) emanating from other cell types, such as macrophages, and remains a topic for further study[55].

To further investigate the role of α4 in the normal development and long-term effect of adipose tissues from birth, we created the constitutive α4 knockout in adipose tissues by using Adiponectin-Cre mice (Aα4KO). Aα4KO mice show a severe absence of WAT and BAT from birth to adulthood under both CD and HFD conditions. We also find that the α4-mediated regulation of insulin signaling is essential not only for the survival of mature adipocytes, but also for embryonic adipogenesis as well as adipocyte regeneration in adults. Because α4 may have broad effects, further studies will be needed to fully define its molecular mechanisms of action. Additionally, the lipodystrophic syndrome observed in Aα4KO mice is similar to the metabolic syndrome associated with generalized lipodystrophy in humans and mice with adipocyte-specific deletion of IR and IGF1R. In all three, there are very low levels of leptin, marked insulin resistance, hyperlipidemia, and fatty liver disease[4,56]. In all, the lipodystrophic phenotype is also associated with dramatically elevated insulin levels and massive

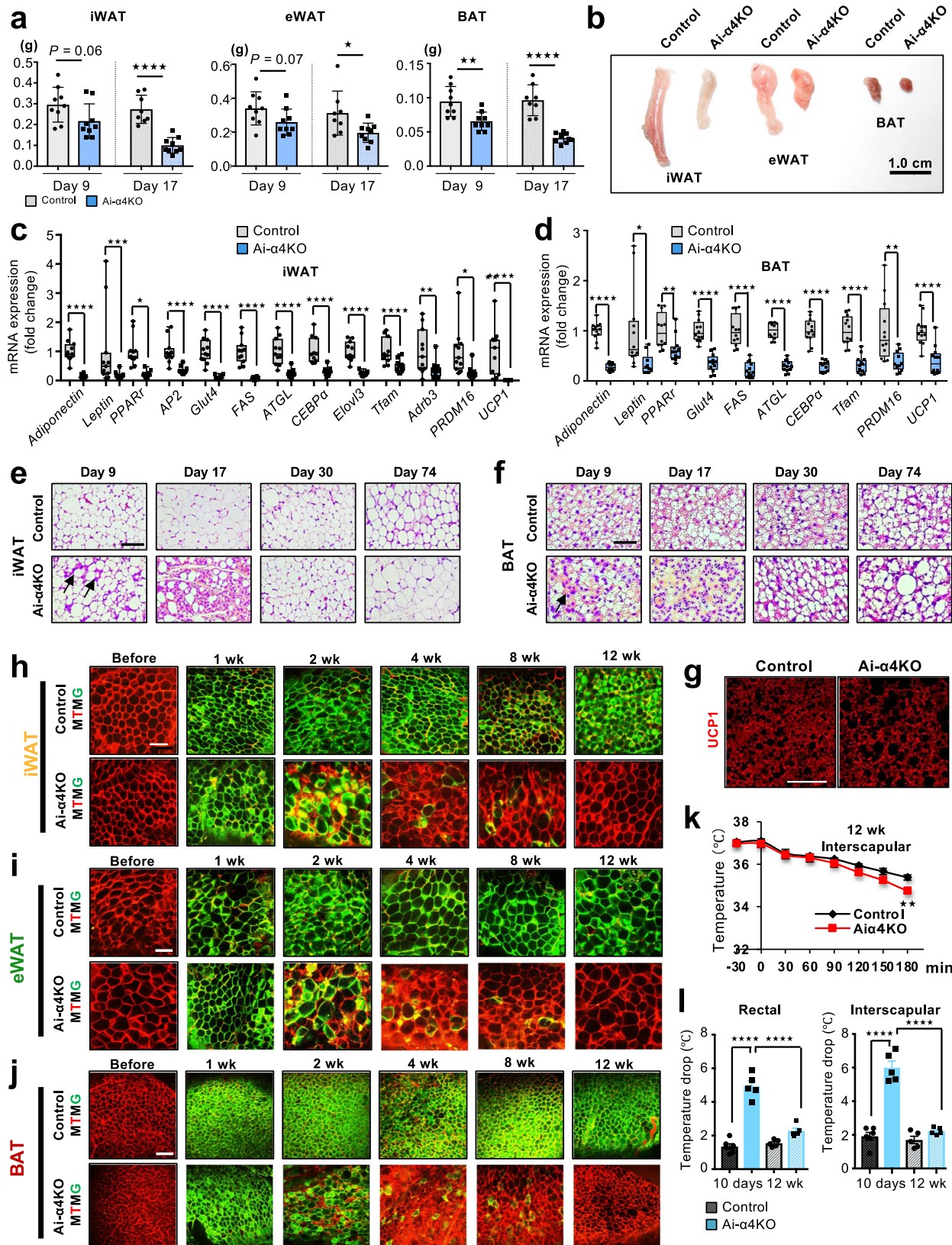

beta-cell hyperplasia, as well as ectopic lipid accumulation in the liver with NAFLD, which can progress to NASH with time or HFD feeding. Limitation in this study is that only male mice were analyzed and generalizability to female mice is not known.

Our study reveals that α4 through its interaction with PP2A and YBX1 has the functional linkage between the Ser/Thr phosphorylation of IR-downstream signaling and membrane protein IR Tyr dephosphorylation, which is essential for resetting the phosphorylation status of IR on the cell membrane. Thus, α4 functions as a crucial regulator of IR signaling via the suppression of YBX1 activation and PTP1B expression. This regulatory feedback loop of insulin signaling is essential for adipogenesis and in the maintenance of WAT and BAT

**Fig. 6 | Adipocyte metabolic dynamics in Ai-α4KO mice. a** Tissue weights (g) of iWAT, eWAT, and BAT from Control and Ai-α4KO on days 9 and 17 after treatment with tamoxifen (Two-tailed Student's $t$-test: $n = 8$/group, $*p = 0.02$, $**p = 0.001$, $***p < 0.0001$). **b** Representative pictures of the adipose tissues of Control and Ai-α4KO on day 17. Scale bars = 1 cm. **c** mRNA expression levels in iWAT from Control ($n = 11$) and Ai-α4KO ($n = 12$) mice on day 17. Box plots are defined in terms of minima and maxima by whiskers, and the center and bounds of box by quartiles (Two-tailed Student's $t$-test, $*p < 0.05$; $**p < 0.01$; $***p < 0.001$). **d** mRNA abundance in BAT ($n = 12$/group) on day 17. Box plots are defined in terms of minima and maxima by whiskers, and the center and bounds of box by quartiles (Two-tailed Student's $t$-test, $*p < 0.05$; $**p < 0.01$; $***p < 0.001$). **e** Results of HE staining of iWAT sections on days 9, 17, 30, and 74. The arrow shows crown-like structures. Scale bars = 100 μm. **f** Results of HE staining of BAT sections from Control and Ai-α4KO on days 9, 17, 30, and 74. The arrow shows crown-like structures. Experiments in e–f were repeated in at least three independent experiments. Scale bars = 50 μm.

**g** UCP1 expression in BAT from Control and Ai-α4KO on days 74. The experiments were repeated independently three times. Scale bars = 50 μm. **h** mTmG lineage tracing of adipocytes in iWAT from Control and Ai-α4KO mice carrying a Rosa-mTmG transgene as shown in Supplementary Fig. 6a before and at wks 1, 2, 4, 8, and 12 after tamoxifen treatment. Scale bars = 100 μm. (**i**) mTmG lineage tracing of adipocytes in eWAT. **j** mTmG lineage tracing of adipocytes in BAT. Experiments in **h**–**j** were repeated in at least three independent experiments. **k** Interscapular temperature in male Control and Ai-α4KO at 30-min intervals during the 3-h exposure of mice 12 wks after tamoxifen administration to an environment at 4 °C (Two-tailed Student's $t$-test, $**p = 0.005$: $n = 5$/group). **l** Comparison of rectal and interscapular temperature drop in Control and Ai-α4KO after 3 h of the 4 °C challenge on day 10 (Control ($n = 6$) and Ai-α4KO ($n = 5$)) with 12 wks after tamoxifen administration ($n = 5$/group) mice (One-way ANOVA post hoc Bonferroni test, $****p < 0.001$). Data are represented as mean ± SEM. Source data are provided as a Source Data file.

function in restoring metabolic homeostasis and preventing systemic metabolic diseases.

## Methods

All experiments were performed in accordance with institutional ethical guidelines and approved by the licensing committee of Kumamoto University (Approval Numbers: A30-051 and 2020-099).

### Mice

Mice were housed at 20–22 °C on a 12 h-light/dark cycle with average 50% Humidity and fed either a CD (#Rodent Diet CE-2, CLEA, Tokyo, Japan) or HFD (60% calories from fat; #HFD32, CLEA, Tokyo, Japan) in the animal facility at Kumamoto University, Japan. All experiments with research animals were performed in accordance with institutional ethical guidelines and approved by the licensing committee of Kumamoto University (Approval Numbers: A30-051 and 2020-099).

Male mice were used for all studies. Adiponectin-Cre mice (C57Bl/6J, 12-week-old, male) were a generous gift from Evan D Rosen (Beth Israel Deaconess Medical Center and Harvard Medical School, Boston, USA) and can now be purchased at Jackson Laboratories (stock no. 02820). Adiponectin-CreER[T2] mice (stock no. 025124, C57Bl/6J, 10-week-old, male) were purchased from Jackson Laboratories. α4 mice have been generated by us (Kumamoto University, CARD, ID: 85)[33]. Control and fat-specific inducible α4 KO (Ai-α4KO) mice were maintained on a mixed (C57Bl/6) background by breeding Adiponectin-CreER[T2] and α4[f/f] mice. For induction of recombination, mice were treated with 100 mg/kg tamoxifen (Sigma) dissolved in 10% ethanol and 90% peanut oil (Sigma) by intraperitoneal injection five times over a 6-day period starting at 2 months of age. In these experiments, tamoxifen was given to all animal groups including control mice, which carried the *floxed alleles*, but lacked the Adiponectin-CreER[T2] transgene. For lineage tracing system of adipocytes Rosa-mTmG (Jax no. 007676, C57Bl/6J, 10-week-old, male) were purchased from Jackson Laboratories and bred to Adiponectin-Cre ER[T2] and Ai-α4KO. The adipose depots were fixed in 10% formalin. Whole mount 3D (Z-stack) imaging was performed using a 2-photon microscope (Olympus FV1000) optimized for td-Tomato and eGFP expression.

### Brown preadipocytes isolation and cell culture

Immortalized mouse brown preadipocytes (WT-1) were derived from the stromal vascular fraction (SVF) of interscapular brown adipose tissue of newborn mice. The SVF cells were immortalized by SV40 T overexpression[34]. Immortalized human brown brown preadipocytes (A41 hBAT-SVF) were derived from the SVF of deep neck fat collected from a human subject[35]. The SVF cells were immortalized by hTert overexpression. The immortalized mouse brown preadipocytes (WT-1) have been deposited to Millipore Sigma and were authenticated by Millipore Sigma (#SCC255). The immortalized human brown

preadipocytes (A41 hBAT-SVF) have been deposited to ATCC and were authenticated by ATCC (#CRL-3385).

Brown preadipocytes were isolated from newborn α4-flox mice by collagenase digestion of interscapular brown fat. The preadipocytes were infected with adenovirus containing GFP alone (to generate control cell line) or GFP-tagged Cre recombinase (to generate α4 knockout preadipocytes). These cells were expanded in DMEM supplemented with 10% heat-inactivated fetal bovine serum (FBS, Sigma), 100 U/ml penicillin and 100 μg/ml streptomycin (Gibco) at 37 °C in a 5% $CO_2$ incubator.

Lentiviral infection. Short hairpin RNA knockdown of α4 was achieved in mouse brown preadipocyte cell line (WT-1) by lentiviral infection. Plates (15 cm) of 70% confluent Lenti-X 293T (Takara, cat. no. #632180) cells were transiently transfected with lentiviral α4 shRNA vector (GeneCopoeia Catalog #: MSH094055-LVRU6GH, target sequence: 5′gcactaagaaatggatctata3′) or scrambled control vector (GeneCopoeia Catalog #: CSHCTR001-LVRU6GH, target sequence: 5′ gcttcgcgccgtagtctta3′) and viral packaging vectors psPax2 and pMD2.G using Fugene HD Transfection Reagent (Promega) according to the manufacturer's instructions. Overexpression of α4 in pLV (VectorBuilder, cat. no. VB190411-1555nef) was also achieved in mouse or human brown preadipocyte cell line by lentiviral infection. Forty-eight hours after transfection, virus-containing medium was collected and passed through a 0.45 μm syringe filter. Polybrene (hexadimethrine bromide; 2 μg/ml) was added, and the medium was applied to proliferating (40% confluency) cells. Twenty-four hours after infection, cells were treated with trypsin and re-plated in a medium supplemented with hygromycin or puromycin. Cells were maintained in DMEM supplemented with 10% FBS, 100 U/ml penicillin and 100 μg/ml streptomycin (Gibco), and cultured at 37 °C in a humidified atmosphere of 5% $CO_2$.

Mouse brown preadipocyte differentiation was induced with an induction mixture containing 20 nM insulin and 1 nM triiodothyronine, 0.5 mM isobutylmethylxanthine, 1 μM dexamethasone, and 0.125 mM indomethacin for 48 h[4,57]. Lipid accumulation was visualized at day 7 by Oil Red O staining. For human brown adipocyte differentiation, we have optimized the insulin concentration (0.5 μM), which is higher than that used for differentiation of mouse brown preadipocytes, as previously described before[35,58]. Cells were grown to confluence and induced with induction mixture containing 0.5 mM isobutylmethylxanthine, 0.1 μM dexamethasone, 0.5 μM human insulin (Sigma-Aldrich, Dallas, TX), 2 nM T3, 30 μM indomethacin, 17 μM pantothenate, 33 μM biotin for another 12 day[35]. Induction medium was changed every 3 days until cells were collected. For the luciferase assay, HEK293 cell line was obtained from RIKEN BRC (RCB1637).

### Metabolic studies

Intraperitoneal glucose tolerance tests (IPGTT) (2.0 g/ kg bw) were performed in unrestrained conscious mice fasted for 6 h. Insulin

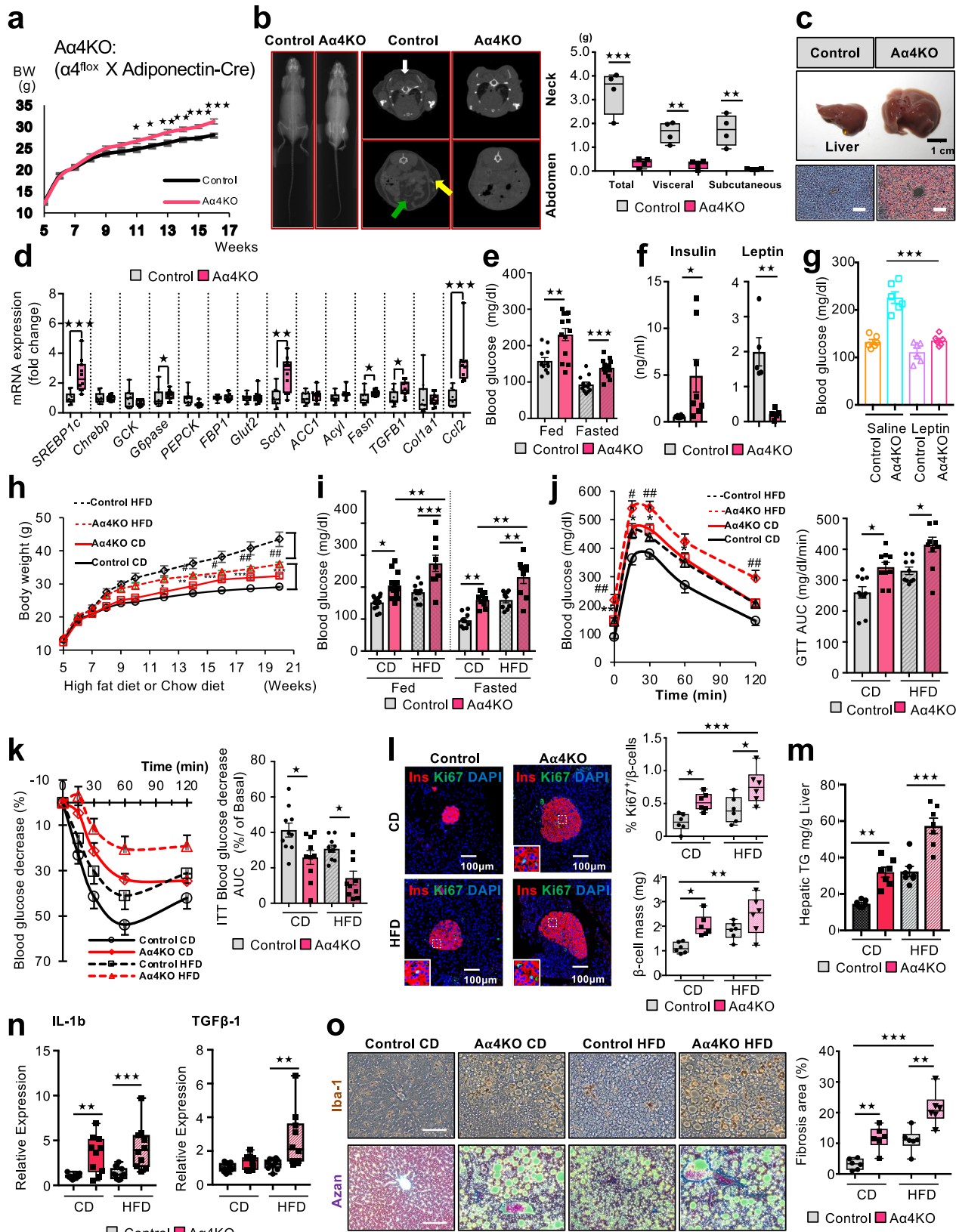

Nature Communications | (2022)13:6092                                                                                                                        14

tolerance tests (1 unit/kg bw, Human R, Lilly) were performed after a 6 h fast. Glucose levels were measured in tail vein blood using Glutest Sensor Neo (Panasonic).

Whole-body energy expenditure was measured at ambient temperature (-22 °C) using a metabolic chamber (model MK-500RQ/MS; Muromachi Kikai, Tokyo, Japan). Mice were housed with alternating

12-h periods of light and dark and were maintained on food and water ad libitum. Mice were adapted to the monitoring for more than 24 h before undergoing a 24-h recording period.

Five-week-old male Control and Aα4KO mice were fed HFD (60% calories from fat; #HFD32, CLEA, Tokyo, Japan). To assess insulin signaling in vivo, 5 U insulin (Sigma-Aldrich) was injected via the inferior

**Fig. 7 | Vulnerability of continued α4-deficient adipose tissues to the development of diabetes and progressive NAFLD owing to a HFD. a** Body weights of 5–16 wk-old male Control and Aα4KO mice fed a CD (Two-tailed Student's *t*-test, *$p < 0.05$; **$p < 0.01$; ***$p < 0.001$: $n = 9$/group). **b** Representative Micro-CT scan images displaying body fat distribution in Control and Aα4KO male mice at 3 months old. The white, yellow, and green arrows indicate the location of interscapular BAT, subcutaneous WAT, and visceral WAT, respectively (left). Total, Visceral, and subcutaneous adipose tissue weights calculated based on the micro-CT images of Control ($n = 4$) and Aα4KO ($n = 4$) mice at 3 months old (right). Box plots are defined in terms of minima and maxima by whiskers, and the center and bounds of box by quartiles (Two-tailed Student's *t*-test, Total: ***$p = 0.0007$; Visceral: **$p = 0.001$; Subcutaneous: **$p = 0.002$). **c** Representative pictures of the livers of Control and Aα4KO mice at 3 months old. Scale bars = 1 cm (upper). Liver sections from Control and Aα4KO mice fed a CD stained with Oil Red O. Scale bars = 100 μm (bottom). **d** mRNA expression levels of genes involved in de novo lipogenesis, inflammation, and fibrosis, as well as those of gluconeogenic enzymes in CD-fed Control and Aα4KO mice at 4 months old. Box plots are defined in terms of minima and maxima by whiskers, and the center and bounds of box by quartiles (Two-tailed Student's *t*-test, *Srebp1c*: ***$p = 0.0007$, *G6pase*: *$p = 0.01$, *Scd1*: **$p = 0.001$, *Fasn*: *$p = 0.04$, *Tgfβ1*: *$p = 0.01$, *Ccl2*: ***$p = 0.0002$: $n = 10$/group). **e** Blood glucose levels corresponding to Control ($n = 12$) and Aα4KO ($n = 13$) mice under fed or fasted at 4 months old. Data are presented as mean ± SEM (Two-tailed Student's *t*-test, **$p = 0.001$, ***$p < 0.0001$). **f** Serum insulin levels corresponding to Control ($n = 8$) and Aα4KO ($n = 8$) mice under fasting (left). Serum leptin levels corresponding to Control ($n = 5$) and Aα4KO ($n = 5$) mice under fasting at 4 months old (right) (Two-tailed Student's *t*-test, *$p = 0.02$, **$p = 0.002$). **g** Blood glucose levels corresponding to random-fed 12-wk-old Control and Aα4KO mice during 2 wks of leptin (10 μg/mouse/day) or saline treatment using Alzet osmotic minipumps (One-way ANOVA post hoc Bonferroni test, ***$p < 0.0001$: $n = 6$/ group).

**h** Body weights of CD-fed and HFD-fed Control and Aα4KO mice for another 16 wks (Two-tailed Student's *t*-test, *$p < 0.05$; **$p < 0.01$; ***$p < 0.001$; #$p < 0.05$; ##$p < 0.01$: Control CD, $n = 9$; Aα4KO CD, $n = 11$; Control HFD, $n = 9$; and Aα4KO HFD, $n = 9$). **(i)** Blood glucose levels of Control and Aα4KO mice under fed (Control CD, $n = 12$; Aα4KO CD, $n = 13$; Control HFD, $n = 11$; and Aα4KO HFD, $n = 9$) and fasted ($n = 9$/group, *$p < 0.05$; **$p < 0.01$; ***$p < 0.001$) (one-way ANOVA post hoc Bonferroni test,). **j** GTT (left) and GTT AUC corresponding to CD- or HFD-fed Control ($n = 10$) and Aα4KO ($n = 10$) after 14 wks of HFD (one-way ANOVA post hoc Bonferroni test, *$p < 0.05$; **$p < 0.01$; ***$p < 0.001$; #$p < 0.05$; ##$p < 0.01$). **k** ITT (left) and the decrease in AUC (right) after 15 wks of HFD (One-way ANOVA post hoc Bonferroni test, CD: *$p = 0.02$, HFD: *$p = 0.01$, $n = 10$/group). **(l)** Pancreatic sections from CD- or HFD-fed Control and Aα4KO immunostained for insulin and Ki67 after 16 wks of HFD. Scale bars = 100 μm. Quantification of Ki67 + insulin + cells in pancreas sections from Control and Aα4KO (left). Quantitation of β-cell mass (right) from Control and Aα4KO (One-way ANOVA followed by Turkey's multiple comparisons, *$p < 0.05$; **$p < 0.01$; ***$p < 0.001$, $n = 6$/group). Box plots are defined in terms of minima and maxima by whiskers, and the center and bounds of box by quartiles. **m** TG content in livers from CD ($n = 7$/group) or HFD ($n = 7$/group) fed mice after 16 wks of HFD (One-way ANOVA post hoc Bonferroni test, **$p = 0.001$, ***$p < 0.0001$). **n** mRNA expression levels of genes involved in inflammation and fibrosis after 16 wks of HFD ($n = 9$/group) or CD ($n = 10$/group). Box plots are defined in terms of minima and maxima by whiskers, and the center and bounds of box by quartiles (one-way ANOVA post hoc Bonferroni test, IL-1β: **$p = 0.004$, ***$p = 0.0008$, *Tgfβ1*: **$p = 0.001$). **o** Liver sections CD- or HFD-fed Control and Aα4KO immunostained for Iba-1 (upper). Representative Azan staining of liver samples (bottom). Scale bars = 100 μm. Quantification of Azan-positive/total area. Box plots are defined in terms of minima and maxima by whiskers, and the center and bounds of box by quartiles (one-way ANOVA post hoc Bonferroni test, *$p < 0.05$; **$p < 0.01$; ***$p < 0.001$, $n = 6$ per group) (right). Source data are provided as a Source Data file.

vena cava. Insulin (ALPCO, cat. no. 80-INSMSU-E01) and leptin (BioVendor, cat. no. RD291001200R) were measured by ELISA.

## Histopathology and immunohistochemistry

Tissues were fixed in 10% formalin, subjected to paraffin-embedded sectioning and stained with hematoxylin or immunostained with anti-UCP1 (Abcam, ab10983, 1:100) and anti-perilipin A (Abcam, ab61682. 1:100) antibodies followed by incubation with the secondary antibody conjugated with Alexa 488 (Invitrogen, A32814, 1:500) and 594 (Invitrogen, A32754, 1:500). TUNEL (terminal deoxynucleotidyl transferase dUTP-mediated nick end labeling) (Takara, cat. no. MK500), cleaved caspase-3 (Cell signaling, #9661, 1:100), Iba-1 (Wako, 019-19741, 1:200), and F4/80 (Serotec, MCA497R, 1:200) immunostaining were performed per manufacturer's instructions. To assess hepatic steatosis, multiple slides of frozen liver sections fixed with 10% buffered formalin for 30 min at room temperature were stained for 7 min with a filtered solution of 0.7% Oil Red O in propylene glycol, and counterstained with hematoxylin for 1 min, washed 3 times with distilled water, and then visualized. Liver sections were also stained with azan to identify fibrosis. To estimate adipocyte size, tissues (iWAT and eWAT) were fixed overnight with neutral-buffered 10% formalin at 4 °C, paraffin embedded, sectioned, and stained with hematoxylin and eosin. Adipocyte area was determined by quantifying the adipocyte area of a total of 20000- cells from 7–9 animals per study group using Adiposoft image analysis software.

## Transmission electron microscopy

Tissues and cells were fixed in 0.1 M phosphate buffer containing 2.5% glutaraldehyde and 2% paraformaldehyde at room temperature and post-fixed with 2% osmium tetroxide solution on ice for 30 min, then processed in a standard manner and embedded in epoxy resin. Semithin sections were cut at 500 nm and stained with 1% toluidine blue to evaluate the quality of preservation. Ultrathin sections (65 nm) were cut on an ultramicrotome (UC7i, Leica), and stained with uranyl acetate and fresh Reynolds' lead citrate at room temperature. Stained grids

were examined with a transmission electron microscope (HT7700, 80kv, Hitachi, Japan).

## Cell sorting

SVF was obtained from iWAT and BAT by treatment with 2 mg/ml collagenase (Sigma) for 45 min at 37 °C. The isolated SVF was resuspended in cold Hank's balanced salt solution (HBSS) with 2% fetal bovine serum (FBS). Cells were incubated with CD45-PerCP-Cy5.5, Clone: 30F-11 (BioLegend, 103131, 1:100), F4/80-APC-Cy7, Clone: BM8 (BioLegend, BL123117, 1:100), CD206-Alex647, Clone: MR5D3 (Bio Rad, MCA2235A647T, 1:50) and CD11c-PE, Clone: HL3 (BD Pharmingen, 553802, 1:100) antibodies for 30 min in HBSS containing 2% FBS on ice and then washed and resuspended in solution with Sytox Blue (Thermo Scientific, S34857, 1:1000). Cells were analyzed on a SH8005 (SONY) cell sorter after selection by forward scatter and side scatter, followed by exclusion of dead cells with Sytox Blue staining, and analyzed for cell-surface markers in FlowJo software (Tree Star). M1 macrophages were identified as F4/80+/CD11c+/CD206− cells. The data are shown as the percentage of total and M1 macrophages.

## Tissue triglyceride quantification

Liver samples (100 mg) were homogenized in 1 ml Folch solution (2:1 v/v chloroform/methanol) and centrifuged at $21,880 \times g$ for 15 min. The supernatants were collected, and TG content determined using Triglyceride Quantification kit (Abnova).

## Ex vivo lipolysis assay in adipose tissue

Lipolysis was assessed as previously described[4]. Inguinal and epididymal fat depots were surgically removed from 2-month-old control and Ai-α4KO at 6 days after the last injection of tamoxifen. After washing with cold PBS, a fragment of 20–30 mg was further cut into five or six pieces and incubated for 2 h at 37 °C in 200 μl of DMEM containing 2% fatty acid-free BSA (Sigma-Aldrich) in the presence or absence of 10 mM isoproterenol (Sigma-Aldrich). Fatty acids released into the medium were quantified using a free fatty acid assay kit (Abcam, cat. no. ab65341) and normalized to the weight of each fat pad.

## Body temperature and cold exposure

Rectal body temperatures of mice were measured using a RET-3 rectal probe (Physitemp). Interscapular body temperatures were assessed using implantable electronic ID transponders (IPTT-300: Bio Medic Data Systems, Inc). For cold exposure, mice were housed individually and subjected to a cold room (ambient temperature 4 °C) without access to food or water. The rectal and interscapular temperature was assessed at the indicated time before and after exposure to the cold, then mice were re-warmed using a heating pad, and all of the mice recovered and were healthy. Skin temperature over BAT was also measured using a thermal imaging camera (FLIR E8 Infrared Camera) and acquired using appropriate software (FLIR Tools Software) without anesthesia.

## Administration of leptin by micro-osmotic pump implantation

8-week-old mice were given a dose of 10 µg/day of recombinant mouse leptin (Sigma) dissolved in sterile saline and administered via Alzet mini-osmotic pumps (DURECT Corp.) designed for 2 weeks' infusion. Pumps were implanted subcutaneously. Saline-filled pumps were implanted in the Control groups.

## Insulin signaling

Cells were serum starved for 3 h with DMEM containing 0.1% BSA and stimulated with 100 nM insulin for indicated times. After stimulation, cells were washed immediately with ice-cold PBS once before lysis and scraped down in RIPA buffer (Millipore, 20-188) supplemented with phosphatase inhibitor and protease inhibitor cocktail (Bimake). Protein concentrations were determined using the Pierce 660 nm Protein Assay Reagent (Bio-Rad). Lysates (10 to 20 µg) were resolved on SDS-PAGE gels, transferred to PVDF membrane for immunoblotting.

## EdU incorporation assay

Cells grown on glass coverslips were fixed with 4% formaldehyde for 15 min at room temperature, rinsed three times in PBS containing 0.3% Triton X for 5 min and blocked in 5% BSA for 30 min at room temperature. For EdU incorporation assay, cells were treated with 10 µM EdU and stained with Click-iT Plus EdU Alexa Fluor 594 Imaging Kit (Thermo Fisher, cat. no. C10639). Images were acquired with a microscope (BZ-9000). The percentage of EUU/DAPI double $^+$ Cells was counted using the BZ-II Dynamic Cell Count Ver. 1.01 program in the BZ-9000 Analysis Software.

## Luciferase assay

The reporter plasmid *Ptp1b* -Luc containing a 2.0 kb fragment of the 5′ flanking region (−2K/+145) of the PTP1B gene, was constructed after PCR amplification of genomic DNA as reported by Fukada et al.[36] and inserted into the PGL4 vector (Promega, cat. no. E6691). The expression plasmids of α4 and YBX1 were constructed in pCMV vectors. Cells were transfected using lipofectamine 3000 (Invitrogen) in 96-well plates. After 24 h, the luciferase assay was performed according to the manufacturer's protocols (Promega, cat. no. E1500).

## Immunoblotting

Membranes were blocked in Starting Block T20 (ThermoFisher) at room temperature for 1 h, incubated with the indicated primary antibody in Starting Block T20 solution overnight at 4 °C. Antibodies against phospho-IR/IGF1R (#3024, 1:1000), IRβ (#3025, 1:1000), phospho-ERK1/2 (T202/Y204) (#9101, 1:1000), ERK1/2 (#9102, 1:1000), phospho-Akt (S473) (#9271, 1:1000), Akt (#4685, 1:1000), phospho-S6 (S235/236) (#2211, 1:2000), S6 (#2317, 1:1000), phospho-YBX1 (S102) (#2900, 1:2000), YBX1 (#4202, 1:1000), α4 (#5699, 1:1000) and GAPDH (#5174, 1:1000) antibodies were from Cell signaling. PTP1B (EPR244207, 1:1000) and Mid2, (ab14749, 1:1000) antibodies were from Abcam. Flag, Clone: M1, (F3040, 1:5000) antibody was from Sigma. PP6 (E-2) (sc-393294, 1:1000) was from Santa Cruz

Biotechnology. Membranes were washed three times with 1X PBST, incubated with HRP-conjugated secondary antibody in Starting Block T20 for 1 h and signals were detected using Immobilon Western Chemiluminescent HRP Substrate (Millipore). Anti-rabbit IgG, HRP-linked Antibody, Cell signaling, (#7074, 1:2000), anti-mouse IgG, HRP-linked Antibody, Cell signaling, (#7076, 1:2000), mouse anti-goat IgG-HRP, Santa Cruz, (sc-2354, 1:1000). All uncropped blots are provided in associated 'source data' file.

## β-cell histology and proliferation

The methods used for analyses have been described previously[59]. Briefly, for quantification of β-cell proliferation paraffin-embedded pancreas sections were co-immunostained with Ki67 (DAKO, M7240, 1:100) and insulin (Abcam, ab7842, 1:500) followed by incubation with the secondary antibody conjugated with Alexa 488 (Jackson Immu-noResearch, 715-546-150, 1:500) and 594 (Invitrogen, A32754, 1:500). Images were recorded using a fluorescence microscope at 20x magnification and cell counting was performed using Image J software. For each sample, at least 1000- 3000 β cells (insulin-positive cells) were counted in a blinded manner by a single person, and the number of Ki67-positive cells were recorded. The percentage of Ki67 positive cells was calculated using the formula: (number of Ki67/insulin double$^+$ cells/total number of insulin$^+$ cells) × 100%. For measurement of β-cell mass, mouse pancreas tissue was fixed in formaldehyde and embedded in paraffin. Five-micron sections of pancreas were cut and immunostained using insulin antibody (Abcam), followed by incubation with the secondary antibody conjugated with Alexa 594 (Invitrogen, A32754, 1:500). Images were captured using fluorescence microscope. Each slide was captured under both ×4 and ×20 magnifications for the whole pancreas and individual islets. Images were analyzed using Image J software. The beta-cell mass was calculated by using the formula: (β-cell area/pancreas area) × pancreatic weight (mg).

## Quantitative RT-PCR

Total RNA was isolated from tissues using RNeasy Mini Kit (Qiagen, Hilden, Germany, 74104) and 1 µg of RNA was reverse transcribed using a High Capacity cDNA Reverse Transcription kit (Applied Biosystems) according to the manufacturer's instructions. Real-time PCR was performed using the LightCycler Fast-Start DNA Master Plus SYBR Green (Roche, Basel, Switzerland). Fluorescence was monitored and analyzed in QuantStudio™ 12K Flex Real-Time PCR System (Thermo Fisher) with primers shown in Supplementary Tables 1 and 2. TBP expression was used to normalize gene expression. Amplification of specific transcripts was confirmed by analyzing melting curve profiles at the end of each PCR.

## Immunoprecipitation and liquid chromatography-tandem mass spectrometry (LC-MS/MS)

To examine protein interactions, cells expressing Flag-tagged α4 were lysed in lysis buffer [20 mM Tris-HCl (pH 7.5), 150 mM NaCl, 1 mM β-glycerolphosphate, 1 mM EDTA (pH 8.0), 1 mM Na$_3$VO$_4$, 1% NP-40, 1× protease inhibitor cocktail] and then centrifuged for 10 min at 21,200 × g. Fifteen milligrams protein lysates were incubated with Flag Magnetic Beads (Sigma) in a total volume of 15 ml for 1 h at 4 °C with end-to-end rotation. The protein complex-bound beads were washed three times with Lysis buffer and eluted the bound Flag-tagged protein with Elution buffer [1× TBS buffer (pH 7.5), 1× protease inhibitor cocktail (Sigma) and 150 ng/µl Flag peptide (Sigma)] by end-to-end rotation at 4 °C for 30 min. The samples were filtered by the Amicon Ultra. The eluted sample was added acetone (final acetone concentration 80% v/v) and incubated for 2 h at −20 °C. After removing the supernatant by centrifugation at 15,000 × g for 15 min at 4 °C, the precipitate was redissolved in 0.5% sodium dodecanoate and 100 mM Tris-HCl, pH 8.5 by using a water bath-type sonicator (Bioruptor

UCD-200 SonicBio Corp., Kanagawa, Japan). The pretreatment of shotgun proteome analysis was performed as previously reported[60]. Peptides were directly injected onto a 75 μm × 20 cm, PicoFrit emitter packed in house with 2.7 μm core shell C18 particles at 45 °C and then separated with an 80 min gradient at a flow rate of 100 nl/min using an UltiMate 3000 RSLCnano LC system (Thermo Fisher Scientific, Waltham, MA, USA). Peptides eluting from the column were analyzed on a Q Exactive HF-X (Thermo Fisher Scientific) for overlapping window DIA-MS[60]. MS1 spectra were collected in the range of 495–785 $m/z$ at 30,000 resolution to set an automatic gain control target of 3e6 and maximum injection time of 55. MS2 spectra were collected in the range of more than 200 $m/z$ at 30,000 resolution to set an automatic gain control target of 3e6, maximum injection time of "auto.", and stepped normalized collision energy of 22, 26, and 30%. An isolation width for MS2 was set to 4 $m/z$ and overlapping window patterns in 500–780 $m/z$ were used window placements optimized by Skyline v4.1[61].

## Lipidomics

Samples were maintained at −80 °C until metabolic profiling was performed. Global lipidomic profiling was performed by Metabolon, Inc (Metabolon, Inc, Durham, NC). Instrument variability was determined by calculating the median relative standard deviation (RSD) for the internal standards that were added to each sample prior to injection into the mass spectrometer. Overall process variability was determined by calculating the median RSD for all endogenous metabolites (i.e., non-instrument standards) present in 100% of the Client Matrix samples, which are technical replicates of pooled client samples. Values for instrument and process variability meet Metabolon's acceptance criteria.

Lipids were extracted from the serum samples in the presence of deuterated internal standards using an automated BUME extraction according to the method of Lofgren et al.[62]. Lipids were extracted from the adipose tissue in the presence of deuterated internal standards using a modified Bligh-Dyer extraction method using methanol/water/dichloromethane. The extracts were concentrated under nitrogen and reconstituted in 0.25 mL of 10 mM ammonium acetate dichloromethane:methanol (50:50). The extracts were transferred to inserts and placed in vials for infusion-MS analysis, performed on a Shimadzu LC with nano PEEK tubing and the Sciex SelexIon-5500 QTRAP. The samples were analyzed via both positive and negative mode electrospray. The 5500 QTRAP scan was performed in MRM mode with the total of more than 1100 MRMs. Individual lipid species were quantified by taking the peak area ratios of target compounds and their assigned internal standards, then multiplying by the concentration of internal standard added to the sample. Lipid class concentrations were calculated from the sum of all molecular species within a class, and fatty acid compositions were determined by calculating the proportion of each class comprised by individual fatty acids.

## RNA-Seq processing

RNA was purified from Qiazol homogenate according to the manufacturer's protocol. RNA-Seq analysis was performed by the Takara Bio Inc. RNA-seq libraries were prepared using the TruSeq Stranded mRNA Sample Prep Kit (Takara). RNA-seq libraries were sequenced using 2×150 bp paired-end reads on the NovaSeq6000 instrument and NovaSeq6000 S4 Reagent Kit.

## Seahorse bioanalyzer

A Seahorse XFe96 Flux analyzer (Agilent Technologies) was utilized to measure oxygen consumption rate (OCR) according to manufacturer's protocol. Brown preadipocytes were seeded into XFe 24 cell culture microplates at the density of 10,000 cells per well and induced differentiation with an induction mixture for 7 days. To measure parameters of mitochondrial function, the XF Cell Mito Stress Test (Agilent, cat. no. 103015-100) was utilized by directly measuring the OCR of cells. Cells were given 1 mM pyruvate, 2 mM glutamine and 10 mM glucose in the XF Base medium (143 mM NaCl, 5.4 mM KCl, 1.8 mM $CaCl_2$, 0.8 mM $MgSO_4$, 0.91 mM $NaH_2PO_4$ and 3 mg/L M Phenol Red), followed by sequential addition of 2 μM oligomycin, 2.5 μM FCCP, and 1 μM antimycin. Basal respiration, ATP production, and maximal respiration capacity was calculated with the subtraction of non-mitochondrial respiration. For the quantitation in the bar graphs, data points for all wells across the four time points were averaged. Cells were lysed in 0.1% SDS solution and protein concentrations were measured and used for normalization of OCR values.

## Proteome data analysis

MS files were searched against a mouse spectral library using Scaffold DIA (Proteome Software, Inc., Portland, OR). The spectral library was generated from mouse protein sequence database (UniProt id UP000000589, reviewed, canonical) by Prosit[63]. The Scaffold DIA search parameters were as follows: experimental data search enzyme, trypsin; maximum missed cleavage sites, 1; precursor mass tolerance, 8 ppm; fragment mass tolerance, 10 ppm; static modification, cysteine carbamidomethylation. The protein identification threshold was set both peptide and protein false discovery rates of less than 1%. Peptide quantification was calculated by EncyclopeDIA algorithm[64] in Scaffold DIA. For each peptide, the four highest-quality fragment ions were selected for quantitation. Protein quantification was estimated from the summed peptide quantification.

## Statistics

Data except for box plots are presented as mean ± SEM and analyzed by two-tailed Student's $t$-test or One-way ANOVA followed by Tukey's or Bonferroni's post hoc tests. Box plots are defined in terms of minima and maxima by whiskers, and the center and bounds of box by quartiles. Biological replicates are indicated in the figure legends. Statistics were carried out using GraphPad 7 or Excel version 16.64. Results were considered significant if $P < 0.05$.

## Reporting summary

Further information on research design is available in the Nature Research Reporting Summary linked to this article.

## Data availability

All data generated from this study are available within the paper and its Supplementary Information/Source data file. Source data are provided with this paper. RNA-seq data generated in this study are available at NCBI GEO database under the accession number GSE210407. Lipidomics raw data have been deposited via the figshare repository [https://doi.org/10.6084/m9.figshare.20417667.v1]. All raw proteomic data have been deposited to the ProteomeXchange Consortium via the PRIDE partner repository with the dataset identifier PXD036915. Source data are provided with this paper.

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

## Acknowledgements

We would like to thank T. Motoyoshi (K.I. Stainer) and Y. Takahashi from the International core-facility of advanced life science at Kumamoto University for excellent technical assistance. We are grateful to mem-bers of the Center for Animal Resources and Development in Kumamoto University for their important contributions to the experiments. This work was supported by Japan Society for the Promotion of Science KAKENHI Grant Numbers JP 18K16208, JP 21K08532, the grant for Basic Research of the Japan Diabetes Society (2018), a grant from Japan Diabetes Foundation, Nippon Boehringer Ingelheim Co., Ltd and Eli Lilly Japan, K.K. (2018), a grant from the MSD Life Science Foundation (2019), a grant from the Manpei Suzuki Diabetes Foundation (2022), a grant from the Takeda Foundation (2019), a grant from the Kanae Foundation (2018), a grant from the Suzuken Memorial Foundation (2018), a grant from the Ono Medical Research Foundation (2018), a grant from the Astellas Foundation for Research on Metabolic Disorders (2018), a grant from the Mochida Memorial Foundation for Medical and Pharmaceutical Research (2018) to M.S. W.C. was supported by NIH grants K01 DK120740.

## Author contributions

M.S. designed and performed the experiments, analyzed data and wrote the paper and supervised the project. S.O., Y.Ok., Y.Ot., K.F., M.I., T.K., Y.S., T.Y., T.F., K. Y., W.C., Y.H.T., N.S., C.R.K. and E.A. helped to perform the experiments.

## Competing interests

The authors declare no competing interests.

## Additional information

[1]Department of Metabolic Medicine, Faculty of Life Sciences, Kumamoto University, 1-1-1 Honjo, Chuoku, Kumamoto 860-8556, Japan. [2]Center for Metabolic Regulation of Healthy Aging (CMHA), Faculty of Life Sciences, Kumamoto University, Kumamoto 860-8556, Japan. [3]Department of Medical Biochemistry, Faculty of Life Sciences, Kumamoto University, 1-1-1 Honjo, Chuoku, Kumamoto 860-8556, Japan. [4]Department of Anatomy and Neurobiology, Faculty of Life Sciences, Kumamoto University, 1-1-1 Honjo, Chuoku, Kumamoto 860-8556, Japan. [5]Department of Biomedical Sciences, New York Institute of Technology College of Osteopathic Medicine, Old Westbury, NY 11568, USA. [6]Sections of Integrative Physiology and Metabolism, Joslin Diabetes Center, Harvard Medical School, Boston, MA 02215, USA. [7]Department of Immunology, Faculty of Life Sciences, Kumamoto University, 1-1-1 Honjo, Chuoku, Kumamoto 860-8556, Japan. [8]Department of Microbiology and Cell Biology, Tokyo Metropolitan Institute of Medical Science, 2-1-6、Kamikitazawa, Setagaya-ku, Tokyo 156-8506, Japan. ✉e-mail: msakaguchi@kumamoto-u.ac.jp

