## [Peer Review File · Nature Communications]

Phosphatase Protector Alpha4 ($\alpha 4$) is Critical for Adipocyte Maintenance and Mitochondrial Homeostasis through Regulation of Insulin SignalingREVIEWER COMMENTS

Reviewer #1 (Remarks to the Author):

Subject: Phosphatase Protector Alpha4 ($\alpha 4$) is Essential for Adipocyte Maintenance and 2 Mitochondrial Homeostasis through Regulation of Insulin Signaling

Sakaguchi et al. report their study on phosphatase binding protein Alpha4 ($\alpha 4$) in insulin signaling and insulin resistance in adipocytes. $\alpha 4$ is essential for the Ser/Thr protein phosphatase PP2A. Alpha4 interacts with the catalytic subunit of PP2A (PP2Ac) and protects the PP2A from the ubiquitin-dependent degradation and maintain the noncanonical PP2A. The author has presented unexpected findings that adipocyte-specific inactivation of $\alpha 4$ impairs insulin-induced Akt-mediated Ser/Thr phosphorylation despite a decrease in the PP2A levels, while reduces insulin-induced insulin receptor Tyr phosphorylation. They demonstrated that decreased association of $\alpha 4$ with Y-box protein 1 (YBX1) led to increased Tyr phosphatase PTP1B expression. In addition, adipocyte-specific knockout of $\alpha 4$ caused impaired adipogenesis and altered mitochondrial oxidation, which resulted in increased inflammation, systemic insulin resistance, hepatosteatosis, islet hyperplasia, and impaired thermogenesis. They concluded that the $\alpha 4$ /YBX1-mediated pathway of insulin receptor signaling is essential for maintaining insulin sensitivity, normal adipose tissue homeostasis and systemic metabolism.

Overall, the study is highly significant and well designed, the conclusions are supported by the substantial amount data, and enough detail provided in the methods for the work to be reproduced. However, there are some concerns needed to be addressed before publication.

1. It is unclear why only brown preadipocytes were used for cell culture studies but not white adipocytes.
2. Line: 146-147: "over the time 6-day time course of "... needs revision
3. Line 347, it is interesting that by Day 30, the widespread adipocyte apoptosis disappeared in Ai- $\alpha 4$ KO mice (Fig. 6e, f), and by Day 74, Ai $\alpha 4$ KO WAT fully recovered, showing a normal morphology, while Ai- $\alpha 4$ KO BAT 350 contained multilocular fat cells, which are typical of BAT as well as large unilocular fat cells that resembled WAT during the recovery period. It will be helpful that the authors provide more explanation of the observed phenomena.
4. Line 391, it is interesting that hepatomegaly with a two-fold increase in liver weight, contributing to the increased body weight observed at 11 weeks of age. It will be helpful that the authors provide more explanation of the observed phenomena.
5. Line 439. it is interesting that despite being protected from apparent obesity, feeding A $\alpha 4$ KO mice 440 with a HFD accelerated diabetes, progressive NAFLD, and pancreatic islet hyperplasia. It will be helpful that the authors provide more explanation of the observed phenomena.
6. Line 596, high concentration of insulin was used for mouse or human brown preadipocyte differentiation for 48 h or 12 days, respectively. It is reported that high insulin treatment for ≥ 48 hours may affect normal insulin signaling and cause insulin resistance in these cells, which may affect the data interpretation. Some discussions on this will be needed.
7. For all WB figures, molecular markers need to be added.
8. Figure 1 I-L, and Figure 2d, Supplementary Figure 3 d, replace $P < 0.001$ or 0.01 with some symbols (*, #) to make better figures. Moreover, effect of insulin is the most interesting result, and it may be better to re-arrange the bar chart as controls $- +$ INS $\alpha 4$ $- +$ INS
9. Supplementary Figure 2. What did " ≥ 3 quantified proteins" mean? How many repeats for this experiment? If it is only once, it should be clearly stated that the $\alpha 4$ interaction partners listed in the figure were identified from $n=1$ experiment in an overexpression system (which is prone to false positives), and further verification will be needed (either bigger n or other method such as Western blotting) to ensure they are $\alpha 4$ binding proteins.

Reviewer #2 (Remarks to the Author):

Phosphatase Protector 1Alpha4 is associated with PP2Ac and is thought to protect the phosphatase from the ubiquitin-dependent degradation. This manuscript investigates the role of Phosphatase Protector 1 Alpha4 on adipocyte function. Using both cell culture models and mouse tissue-specific knockdown approaches, the authors find pleiotropic effects emanating from the knockdown of this protein, which in the in-vivo model results in apoptotic cell death, inflammation and consequent metabolic phenotypes. The authors postulate a specific role for Alpha4 on insulin signaling, and interpret their results as resulting from impairment in the insulin signaling pathway. However, the data point strongly to a less specific, essential role for the protein, which affects multiple signaling pathways and cell viability. Observed signaling and metabolic effects stem from compromised adipocyte viability, rather than from specific modulation of insulin signaling. Specific concerns are as follows:

Figure 1 shows enforced Alpha4 overexpression resulting in an increase in UCP1 expression of ~2-fold. If functionally significant, increased UCP1 levels would be expected to result in changes in uncoupled respiration, but instead the authors find increased FCCP-induced maximal respiratory capacity. This result suggests an increase in mitochondrial capacity but not a functional effect of UCP1. The authors should measure UCP1 protein expression and other mitochondrial markers to clarify the nature of the effect, if any. Please clarify what tests were used to account for multiple comparisons

Knockdown of Alpha4 resulted in decreased levels of adipose differentiation markers, as well as insulin responsiveness as measured by phospho-IRb and pAkt levels. The authors suggest that failure of differentiation is due to decreased insulin signaling, but this inference is not supported by the data. A pleiotropic effect of Alpha4 KO could affect both differentiation and signaling, or multiple other cell pathways. Are cell viability, proliferation rate, metabolic state, etc compromised? Please clarify what tests were used to account for multiple comparisons

Figure 2 attempts to elucidate the mechanisms by which Alpha 4 KO impairs insulin signaling. Using mass spectroscopy the authors identify YBX1 as interacting with flag-tagged Alpha4, and also find increased levels of PTP1B expression. However, the data are correlational and do not directly test the hypothesis that it is YBX1 interactions with Alpha4 functionally control PTP1B levels, not that PTP1B levels result in observed alterations in insulin signaling. The possibility that knockdown of Alpha4 may have pleiotropic actions that independently result in the observed changes. Please clarify whether replicates are distinct cell isolates, or technical replication of three culture wells.

Figure 3 illustrates the effects on inguinal, epididymal and brown adipose tissue of inducible knockdown of Alpha 4 in adipocytes. After 6 days of knockdown, marked alterations in adipose tissue architecture are observed, consistent with inflammation. This is accompanied by changes in the lipidomic composition of the tissue. However, whether these lipidomic effects are occurring within adipocytes, or are due to changes in the cellular composition of the adipose tissue such as that which would ensue in an inflammatory state, is not addressed. It is also notable that the effects are not what would be expected solely from decreased insulin signaling, which would be a global decrease in adipocyte size due to enhanced lipolysis. Indeed, global transcriptomics of the tissue is consistent with an acute inflammatory state which could arise from cell death, which is indeed confirmed in Figure 5.

The findings using a constitutive knockdown of Alpha4 in adipocytes are also consistent with a requirement for Alpha4 for cell viability, as determined by the classical lipodystrophy phenotype of the mice, including insulin resistance and liver steatosis exacerbated by high-fat diet feeding.

Reviewer #3 (Remarks to the Author):

The authors found that adipocyte specific loss of a4 led to impaired insulin signaling as read out by Akt-mediated Ser/Thr phosphorylation the impacts brown adipocyte differentiation and thermogenic capacity. Interestingly a4 is sufficient to drive increased UCP1 expression and increases respiration without other substantial impacts on differentiation. However, KD demonstrates that a4 is necessary for insulin signaling and differentiation.

The impact of a4 on insulin signaling is driven by a4 interaction with YBX1, a transcriptional regulator of PTP1b. The authors demonstrated that a4 directly binds to YBX1 and facilitates phosphorylation. Loss of a4 in the adipocyte alone led to increased apoptosis, increased macrophage infiltration and activation, as well as decreased adipose tissue mass. The loss of a4 in the adipocyte also led to increased hepatic steatosis and inflammation.

Several factors make this manuscript stand out including the excellent use of model organisms, specifically the inducible models to bypass the impact on differentiation that was observed in the KD cells by gene expression as well as the nonbiased approach to molecular phenotyping to assess direct regulation. I recommend the manuscript for publication in Nature Communications with major and minor revisions detailed below. The challenges in the manuscript are the breadth of phenotypes that are present and the focused assessment of a4 mechanistic function.

Major Revisions:

1. In figure 1, the authors observe increased respiration and UCP1 with OE of a4. To be certain these changes are solely due to UCP1, the authors should look at mitochondrial abundance, at least a few more mitochondrial transcripts in the cells such as Cox5b, Cox7a, or Ndufs which are decreased in the RNA-seq analysis in the Ai-a4KO mice. The mitochondrial imaging in the OE would also be helpful in the interpretation of the respiration data.
2. The major mechanistic point is that YBX1 activity is altered with a4 binding. The authors sufficiently show Ybx1 binding to a4, however, the functional readout would be better shown with a luciferase assay with a Ybx1 binding motif. Further demonstration that the Ser residues 102 or 165 are important would be best shown through site directed mutagenesis and then demonstration that a5 OE had no impact on YBX activity.

Minor Revisions:

1. Minor figure corrections:
 - a. Figure 1 i,j,k, and l the -/+ sign for the addition of insulin is misaligned
 - b. Figure 2d Ins is cut off in the x-axis for p-YBX1
 - c. Figure 4 and 5 with the heat maps it may help to cluster some of the pathways with a GO analysis just to show overall pathway changes. The authors do this in the write up but it would help readability since its difficult to read some of the transcript names in the figure.
 - d. 7h x-axis should be labeled in Time on HFD or chow (weeks) in the same font size
2. The resolution of several figures should be improved including: S1C, Figure 6 e&f, Figure 7C.
3. In figure 1, the authors observed OE of a4 led to decreased leptin in primary brown adipocytes, in figure 6 c, and in the Aia4KO mice the blood leptin levels were decreased. Given the regulation of PTP1B by YBX1, is there potentially some type of regulatory loop?
4. The 60% decrease in PP2A in the a4KO cells, while YBX1 phosphorylation is increased and PTP1B is increased. Is there any indication of what is controlling the PP2A levels? Did the authors look at PP2A levels in the BAT of the Ai-a4KO mice? It would add to the significance of the manuscript if these changes were discussed.
5. For the methods section the authors should adhere to the lipidomics standard initiative for blanks, internal standards, normalization method and sharing of raw data <https://lipidomics-standards-initiative.org/>
6. The authors detail several exciting phenotypes in the Aia4KO mice, the discussion should hint at potentials for further exploration. Some striking findings to follow up on are: 1) The infiltration of macrophages and the activation differences in Aia4KO mice. Others including work from the lab of Amira Klip, have shown that macrophages in the adipocyte provide iron and other macronutrients to

the adipocyte which could explain the mitochondrial morphological differences. 2) Another striking phenotype that has been explored is the work of Granneman lab that shows the importance of the macrophages in fat cell turn over and beige adipogenesis.

7. The authors are likely aware of the work from the Scherer group demonstrating that tamoxifen induces apoptosis in adipocytes. I am curious if the authors performed a dose response especially in isolated adipocytes to ensure the apoptotic phenotype isn't due to altered sensitivity to tamoxifen in the control and Aia5KO mice.

Responses to the REVIEWERS' COMMENTS

Reviewer #1 (Remarks to the Author):

Subject: Phosphatase Protector Alpha4 ($\alpha 4$) is Essential for Adipocyte Maintenance and Mitochondrial Homeostasis through Regulation of Insulin Signaling

Sakaguchi et al. report their study on phosphatase binding protein Alpha4 ($\alpha 4$) in insulin signaling and insulin resistance in adipocytes. $\alpha 4$ is essential for the Ser/Thr protein phosphatase PP2A. Alpha4 interacts with the catalytic subunit of PP2A (PP2Ac) and protects the PP2A from the ubiquitin-dependent degradation and maintain the noncanonical PP2A. The author has presented unexpected findings that adipocyte-specific inactivation of $\alpha 4$ impairs insulin-induced Akt-mediated Ser/Thr phosphorylation despite a decrease in the PP2A levels, while reduces insulin-induced insulin receptor Tyr phosphorylation. They demonstrated that decreased association of $\alpha 4$ with Y-box protein 1 (YBX1) led to increased Tyr phosphatase PTP1B expression. In addition, adipocyte-specific knockout of $\alpha 4$ caused impaired adipogenesis and altered mitochondrial oxidation, which resulted in increased inflammation, systemic insulin resistance, hepatosteatosis, islet hyperplasia, and impaired thermogenesis.

They concluded that the $\alpha 4$ /YBX1-mediated pathway of insulin receptor signaling is essential for maintaining insulin sensitivity, normal adipose tissue homeostasis and systemic metabolism.

Overall, the study is highly significant and well designed, the conclusions are supported by the substantial amount data, and enough detail provided in the methods for the work to be reproduced. However, there are some concerns needed to be addressed before publication.

Response

---- Thank you very much for your overall evaluation of our manuscript. Owing to your constructive comments, we think our revised manuscript has been improved significantly.

1. *It is unclear why only brown preadipocytes were used for cell culture studies but not white adipocytes.*

Response

--- $\alpha 4$ mRNA expression in BAT was higher than that of WAT. We now explain this in the text with additional information in Supplementary Figure 1a.

2. *Line: 146-147: “over the time 6-day time course of ”... needs revision*

Response

--- We have corrected the sentence to “during the 6-day culture of”.

3. *Line 347, it is interesting that by Day 30, the widespread adipocyte apoptosis disappeared in Ai- $\alpha 4$ KO mice (Fig. 6e, f), and by Day 74, Ai $\alpha 4$ KO WAT fully recovered, showing a normal morphology, while Ai- $\alpha 4$ KO BAT 350 contained multilocular fat cells, which are typical of BAT as well as large unilocular fat cells that resembled WAT during the recovery period. It will be helpful that the authors provide more explanation of the observed phenomena.*

Response

---Thank you for your suggestion. Additional explanation has been added to the text including additional data in Figure 6g and Supplementary Figure 6a as follows: Both the multilocular and unilocular cells in Ai- $\alpha 4$ KO BAT were positive for UCP-1 protein (Figure 6g). In the revised manuscript, we now discuss this point in more detail in the Discussion section.

4. *Line 391, it is interesting that hepatomegaly with a two-fold increase in liver weight, contributing to the increased body weight observed at 11 weeks of age. It will be helpful that the authors provide more explanation of the observed phenomena.*

Response

--- We have added more explanation in the text as follows.

In response to the severe loss of WAT and BAT, A $\alpha 4$ KO mice showed enlarged livers with an average two-fold increase in liver weight, presumably secondary to uptake and accumulation of circulating lipid contents (Fig,7c). Consequently,

A α 4KO mice developed hepatomegaly associated with increased intrahepatic triglyceride accumulation and hepatocyte balloon degeneration, the former contributing to the increased body weight observed at 11 weeks of age (Fig. 7c and Supplementary Fig. 7b-d).

5. Line 439. it is interesting that despite being protected from apparent obesity, feeding A α 4KO mice 440 with a HFD accelerated diabetes, progressive NAFLD, and pancreatic islet hyperplasia. It will be helpful that the authors provide more explanation of the observed phenomena.

Response

--- In response to your suggestion, we have expanded the explanation of this in the text:

Overall, despite being protected from obesity, feeding A α 4KO mice with a HFD accelerated diabetes, progressive NAFLD, and pancreatic islet hyperplasia. Taken together, the lipodystrophic A α 4KO mice developed liver injury due to effects of lipotoxicity, and HFD feeding exacerbated the liver disease, eventually causing altered whole-body metabolism and overt disease pathogenesis.

6. Line 596, high concentration of insulin was used for mouse or human brown preadipocyte differentiation for 48 h or 12 days, respectively. It is reported that high insulin treatment for \geq 48 hours may affect normal insulin signaling and cause insulin resistance in these cells, which may affect the data interpretation. Some discussions on this will be needed.

Response

--- For the mouse brown adipocyte differentiation *in vitro*, we have been using the induction mixture containing 20 nM insulin, as described before in multiple publications (Tseng and Kahn et al., Nature 2008, Boucher and Kahn et al., Nature Comm. 2012, Sakaguchi and Kahn et al., Cell Metab. 2017). While still superphysiologic, in comparison with the insulin concentrations (~850 nM) used for mouse brown adipocyte differentiation by other groups (Seale and Spiegelman et al. Nature 2008; Kajimura et al., Nature 2008; Kong and Rosen et al., Cell 2014), we think the insulin concentration for mouse differentiation is closer to physiological and appropriate range for the analysis.

In the experiments using human brown adipocyte differentiation, we have used 500 nM insulin as described before (Xue and Tseng et al., Nature Medicine 2015). Although this is higher than the concentration used with mouse brown preadipocytes, differentiated human BAT displayed remarkable ability in glucose uptake by the insulin stimulation (Xue and Tseng et al., Nature Medicine 2015) i.e., no apparent insulin resistance. Furthermore, recently another group tested insulin concentrations from 0 nM to 1720 nM for differentiation of human brown adipocyte (Wang et al., Scientific Report 2018) and found that at concentrations ranging from 172–860 nM, insulin could appreciably promote adipogenesis and brown fat gene expression. Therefore, we considered 500 nM appropriate and effective to induce human brown adipocyte differentiation.

As the reviewer suggested, we agree that the insulin concentrations used for cell differentiation in these *in vitro*-experiments are higher than those of physiological conditions *in vivo*. This is likely because insulin is serving as a stimulus of both the insulin and IGF-1 receptors. We provide believe the above data allow for better interpretation of both the *in vitro* and *in vivo* experimental methods. We have added these references and described this point in the Methods section.

7. For all WB figures, molecular markers need to be added.

Response

---We have added molecular markers in all WB figures.

8. Figure 1 I-L, and Figure 2d, Supplementary Figure 3 d, replace $P < 0.001$ or 0.01 with some symbols (*, #) to make better figures. Moreover, effect of insulin is the most interesting result, and it may be better to re-arrange the bar chart as controls \rightarrow INS alpha4 \rightarrow INS

Response

--- We replaced $P < 0.001$ or 0.01 with some symbols (*, #) to make figures easier to read, including Figure 1 I-L, and Figure 2d, Supplementary Figure 3 d,e. We have also re-arranged the bar chart as controls \rightarrow INS alpha4 \rightarrow INS.

9. Supplementary Figure 2. What did “ ≥ 3 quantified proteins” mean? How many repeats for this experiment? If it is only once, it should be clearly stated that the $\alpha 4$ interaction partners listed in the figure were identified from $n=1$ experiment in

an overexpression system (which is prone to false positives), and further verification will be needed (either bigger n or other method such as Wester blotting) to ensure they are α 4 binding proteins.

Response

---Thank you for your comment and apologize for the confusion. The $n \geq 3$ refers to the number of quantified peptides, not the number of experiments. We selected the candidate molecules from the data, which detected at least three quantified peptides. We have modified the sentence for clarity. Also, for the mass spectroscopic proteomic analysis we used only 1 Control but 2 independent samples for α 4 overexpressing cells.

and further verification will be needed (either bigger n or other method such as Wester blotting) to ensure they are α 4 binding proteins.

Response

---We have also performed IP and western blotting in Figure 2b (n=3) to confirm the interactions. The results clearly demonstrate that PP2A and YBX1 are α 4 binding proteins. The results are included in Figure 2b.

Reviewer #2 (Remarks to the Author):

Phosphatase Protector 1Alpha4 is associated with PP2Ac and is thought to protect the phosphatase from the ubiquitin-dependent degradation. This manuscript investigates the role of Phosphatase Protector 1 Alpha4 on adipocyte function. Using both cell culture models and mouse tissue-specific knockdown approaches, the authors find pleiotropic effects emanating from the knockdown of this protein, which in the in-vivo model results in apoptotic cell death, inflammation and consequent metabolic phenotypes. The authors postulate a specific role for Alpha4 on insulin signaling, and interpret their results as resulting from impairment in the insulin signaling pathway. However, the data point strongly to a less specific, essential role for the protein, which affects multiple signaling pathways and cell viability. Observed signaling and metabolic effects stem from

compromised adipocyte viability, rather than from specific modulation of insulin signaling. Specific concerns are as follows:

Response

---Thank you for raising this possibility. We have now examined the impact of $\alpha 4$ KO mice at the early stage of $\alpha 4$ depletion and shown that this does not impair cell viability in the adipose tissues. Also in response to this question, we now show experiments for insulin signaling that were performed without causing apoptotic cell death and or tissue inflammation (Figure 3f and Supplementary Figure 3f). We appreciate the comments and think that our manuscript has been improved much concerning the molecular mechanism in the regulation of insulin signaling through $\alpha 4$.

Figure 1 shows enforced Alpha4 overexpression resulting in an increase in UCP1 expression of ~2-fold. If functionally significant, increased UCP1 levels would be expected to result in changes in uncoupled respiration, but instead the authors find increased FCCP-induced maximal respiratory capacity. This result suggests an increase in mitochondrial capacity but not a functional effect of UCP1. The authors should measure UCP1 protein expression and other mitochondrial markers to clarify the nature of the effect, if any. Please clarify what tests were used to account for multiple comparisons

Response

---We agree with the reviewer and the request for measurement of other mitochondrial markers as well as UCP1. We now have measured UCP1 protein expression and found that Alpha4-overexpression results in a modest increase in UCP1 expression of 1.4-fold. We also measured other mitochondrial markers and found that Cox5b, Cox7a, and Ndufs were significantly increased in Alpha4-overexpressed cells. These results suggest that the increased FCCP-induced maximal respiratory capacity could be explained by an increase in total mitochondrial capacity rather than a sole effect of UCP1. We now describe multiple statistical comparisons performed by ANOVA followed by the Bonferroni post hoc test, have added new text and these data in Supplementary Figure 1e, f.

Knockdown of Alpha4 resulted in decreased levels of adipose differentiation

markers, as well as insulin responsiveness as measured by phospho-IRb and pAkt levels. The authors suggest that failure of differentiation is due to decreased insulin signaling, but this inference is not supported by the data. A pleiotropic effect of Alpha4 KO could affect both differentiation and signaling, or multiple other cell pathways. Are cell viability, proliferation rate, metabolic state, etc compromised? Please clarify what tests were used to account for multiple comparisons

Response

--- As the reviewer notes, the Ser/Thr phosphatase protein, PP2A, is a ubiquitous eukaryotic enzyme that regulates various cellular processes through the dephosphorylation of many substrates. PP2A functions in many stages of the cell cycle, and its effect is widespread, which makes elucidation of the regulatory mechanism of $\alpha 4$ in detail in different signaling pathways challenging.

We have now added new *in vitro* and *in vivo* experimental data in response to your concern. In this we show that $\alpha 4$ KD cells have similar growth rates and doubling times in culture. $\alpha 4$ KD cells also show no measurable change in DNA synthesis or cell proliferation rate compared with control cells. Thus, any effects of $\alpha 4$ KD on differentiation and signaling occur without affecting cell viability and proliferation *in vitro* (Supplementary Figure 1k-m). In the *in vivo* experiments, we have shown insulin responsiveness is attenuated as early as three days after tamoxifen treatment. At this stage, there was no apparent change in adipocyte diameter and size (Figure 3a-c). We also measured inflammation markers and apoptotic cells by immunohistochemistry staining in both WAT and BAT sections with staining for TUNEL and cleaved-caspase 3 at day-3 after tamoxifen treatment. At this time point, we did not detect any evidence of apoptosis in the adipose tissue, as shown in Figure 3f and Supplementary Figure 3f. In the discussion section, we now explain that our experiments were directed at determining alternations of insulin signaling after $\alpha 4$ depletion, when the cells are viable with normal proliferation potential avoiding any potential effects of cell damage caused by the PP2A alteration. We also describe multiple statistical comparisons performed by ANOVA followed by the Bonferroni post hoc test.

Figure 2 attempts to elucidate the mechanisms by which Alpha 4 KO impairs insulin signaling. Using mass spectroscopy the authors identify YBX1 as interacting with flag-tagged Alpha4, and also find increased levels of PTP1B

expression. However, the data are correlational and do not directly test the hypothesis that it is YBX1 interactions with Alpha4 functionally control PTP1B levels, not that PTP1B levels result in observed alterations in insulin signaling.

The possibility that knockdown of Alpha4 may have pleiotropic actions that independently result in the observed changes. Please clarify whether replicates are distinct cell isolates, or technical replication of three culture wells.

Response

---We agree with the reviewer.

We have now employed a reporter system using a *Ptp1b* promoter-driven luciferase vector as reported by Fukada et al. (EMBO 2003), and analyzed the effect of $\alpha 4$ on PTP1B expression. Strikingly, we found the $\alpha 4$ overexpression significantly reduced the transactivation ability of YBX1 on *Ptp1b* promoter (Figure 2e). These results demonstrate that YBX1 interaction with $\alpha 4$ directly regulates PTP1B expression.

The possibility that knockdown of Alpha4 may have pleiotropic actions that independently result in the observed changes.

Response

---As noted above, we have confirmed that the loss of $\alpha 4$ on insulin signaling occurs before inducing any compromise in adipocyte viability. However, we have indicated in the revised manuscript, that Alpha4 may have broad effects and that further studies will be needed to fully define its molecular mechanisms of action.

Please clarify whether replicates are distinct cell isolates, or technical replication of three culture wells.

Response

---We generated five independent $\alpha 4$ KD clones and carried out the experiments on these clones. For the insulin signaling analysis, three clones were used in three independent experiments. We describe them in the new version.

Figure 3 illustrates the effects on inguinal, epididymal and brown adipose tissue of inducible knockdown of Alpha 4 in adipocytes. After 6 days of knockdown,

marked alterations in adipose tissue architecture are observed, consistent with inflammation. This is accompanied by changes in the lipidomic composition of the tissue. However, whether these lipidomic effects are occurring within adipocytes, or are due to changes in the cellular composition of the adipose tissue such as that which would ensue in an inflammatory state, is not addressed. It is also notable that the effects are not what would be expected solely from decreased insulin signaling, which would be a global decrease in adipocyte size due to enhanced lipolysis. Indeed, global transcriptomics of the tissue is consistent with an acute inflammatory state which could arise from cell death, which is indeed confirmed in Figure 5.

Response

--We thank the reviewer for this comment. As the reviewer suggested, we have now immunostained both WAT and BAT sections for the inflammatory marker F4/80 on day six after tamoxifen treatment. At this point, Ai- α 4KO mutant mice have begun to show some inflammation with the macrophage infiltrations in both WAT and BAT. We have added these data into Supplementary Figure 5 i-k.

“Lipidomic effects” occur not only within adipocytes but also due to changes in the cellular compositions of the adipose tissue, including macrophages. Insulin inhibits apoptosis and lipolysis in mature adipocytes (Sakaguchi et al., Cell Metab. 2017). Thus, we compared *ex vivo* lipolysis with or without isoproterenol stimulation. At the 6-day after tamoxifen treatment, Ai- α 4KO mice showed a significant increase in lipolysis even without stimulation by isoproterenol in both subcutaneous and perigonadal depots. These results demonstrate that α 4 is required to maintain adipose tissues by suppressing lipolysis through insulin and thus inhibiting adipocyte apoptosis. Finally, we have added these data in Figure 3i and Supplementary Figure 3h, and explained this point in detail in the discussion section.

The findings using a constitutive knockdown of Alpha4 in adipocytes are also consistent with a requirement for Alpha4 for cell viability, as determined by the classical lipodystrophy phenotype of the mice, including insulin resistance and liver steatosis exacerbated by high-fat diet feeding.

Response

---Thank you for your evaluation of our manuscript, and the *in vitro* and the *in vivo* experiments.

Reviewer #3 (Remarks to the Author):

The authors found that adipocyte specific loss of a4 led to impaired insulin signaling as read out by Akt-mediated Ser/Thr phosphorylation the impacts brown adipocyte differentiation and thermogenic capacity. Interestingly a4 is sufficient to drive increased UCP1 expression and increases respiration without other substantial impacts on differentiation. However, KD demonstrates that a4 is necessary for insulin signaling and differentiation.

The impact of a4 on insulin signaling is driven by a4 interaction with YBX1, a transcriptional regulator of PTP1b. The authors demonstrated that a4 directly binds to YBX1 and facilitates phosphorylation. Loss of a4 in the adipocyte alone led to increased apoptosis, increased macrophage infiltration and activation, as well as decreased adipose tissue mass. The loss of a4 in the adipocyte also led to increased hepatic steatosis and inflammation.

Several factors make this manuscript stand out including the excellent use of model organisms, specifically the inducible models to bypass the impact on differentiation that was observed in the KD cells by gene expression as well as the nonbiased approach to molecular phenotyping to assess direct regulation. I recommend the manuscript for publication in Nature Communications with major and minor revisions detailed below. The challenges in the manuscript are the breadth of phenotypes that are present and the focused assessment of a4 mechanistic function.

Response

---We appreciate the reviewer's evaluation of our manuscript and their helpful advice. Owing to your comments, we believe our manuscript has improved.

Major Revisions:

1. In figure 1, the authors observe increased respiration and UCP1 with OE of a4. To be certain these changes are solely due to UCP1, the authors should look at

mitochondrial abundance, at least a few more mitochondrial transcripts in the cells such as Cox5b, Cox7a, or Ndufs which are decreased in the RNA-seq analysis in the Ai- α 4KO mice. The mitochondrial imaging in the OE would also be helpful in the interpretation of the respiration data.

Response

---We measured the mitochondrial markers and found Cox5b, Cox7a, and Ndufs were significantly increased in α 4 overexpressed (OE) cells. These results showed the increased FCCP-induced maximal respiratory capacity could be explained by an increase in mitochondrial capacity rather than a sole effect of UCP1. We also performed mitochondrial imaging in the α 4 OE cells. Consistent with the increase of mitochondria transcripts in the α 4 OE cells, the mitochondria are more prominent in size than those of Control cells. We have added these data in Supplementary Figure 1e, f and 1j.

2. The major mechanistic point is that YBX1 activity is altered with α 4 binding. The authors sufficiently show Ybx1 binding to α 4, however, the functional readout would be better shown with a luciferase assay with a Ybx1 binding motif. Further demonstration that the Ser residues 102 or 165 are important would be best shown through site directed mutagenesis and then demonstration that α 5 OE had no impact on YBX activity.

Response

---As the reviewer suggested, we have now employed a reporter system using a *Ptp1b* promoter-driven luciferase vector as reported by Fukada et al. (EMBO 2003), and analyzed the effect of α 4 on PTP1B expression. We found that the α 4 overexpression had a striking and significant effect to reduce the transactivation ability of YBX1 on *Ptp1b* promoter. These results show that YBX1 interaction with α 4 readily regulates PTP1B expression (Fig. 2e).

Minor Revisions:

1. Minor figure corrections:

a. Figure 1 i,j,k, and I the -/+ sign for the addition of insulin is misaligned

Response

---Thank you for your comments. We have modified all of the points in the revised figures.

b. Figure 2d Ins is cut off in the x-axis for p-YBX1

Response

---We modified the Figure as suggested.

c. Figure 4 and 5 with the heat maps it may help to cluster some of the pathways with a GO analysis just to show overall pathway changes. The authors do this in the write up but it would help readability since its difficult to read some of the transcript names in the figure.

Response

---We thank you for the comments on clustering the heatmaps and pathway analysis in the Figures to help readability. We have now clustered the pathway analysis and the heatmaps in Figures 4 and 5. In addition to the KEGG pathway analysis, we also performed a GO analysis in Supplementary Figure 4c and 5a.

d. 7h x-axis should be labeled in Time on HFD or chow (weeks) in the same font size

Response

---Thank you for your careful review, and we followed the suggestion.

2. The resolution of several figures should be improved including: S1C, Figure 6 e&f, Figure 7C.

Response

---We have modified the Figures as suggested.

3. In figure 1, the authors observed OE of a4 led to decreased leptin in primary brown adipocytes, in figure 6 c, and in the Aia4KO mice the blood leptin levels were decreased. Given the regulation of PTP1B by YBX1, is there potentially some type of regulatory look?

Response

---Thank you for the interesting comments. Our experiments showed a controversial phenomenon regarding the leptin mRNA expression between the Ai- α 4KO and α 4OE cells. The decrease in blood leptin levels may be due to a lack of adipocyte differentiation in Ai- α 4KO; on the other hand, α 4OE cells also showed low leptin expression (high expression of Ucp1). This suggests that there may be a regulatory interaction between the α 4 mediated loop in adipocyte differentiation and leptin expression. For example, SP1 is one of the transcription factors controlling leptin expression, and the activated PP2A may enhance SP1 transcriptional activity when PTP1B is overexpressed (Shimizu. et al. JBC 2003), suggesting the existence of PP2A/SP1/PTP1B-dependent regulation of leptin transcription. These are exciting and essential issues will be the subject of future studies.

4. The 60% decrease in PP2A in the α 4KO cells, while YBX1 phosphorylation is increased and PTP1B is increased. Is there any indication of what is controlling the PP2A levels? Did the authors look at PP2A levels in the BAT of the Ai- α 4KO mice? It would add to the significance of the manuscript if these changes were discussed.

Response

--As previously described, a phosphatase protector α binds with a significant amount of PP2Ac and maintains PP2Ac by protecting it from ubiquitin-dependent degradation (Kong et al. Mol Cell 2009). We also confirmed that α 4 interacts with PP2Ac in brown preadipocytes; thus, α 4 also regulates PP2Ac levels in brown preadipocytes. As suggested by the reviewer, we also quantified the PP2Ac levels in both the BAT and WAT of the Ai- α 4KO. These data strengthened our results. We have added the additional results in the manuscript (Figure 3 d,e and Supplementary Figure 3d, e).

5. For the methods section the authors should adhere to the lipidomics standard initiative for blanks, internal standards, normalization method and sharing of raw data <https://lipidomics-standards-initiative.org/>

Response

---Thank you for the comment. Global lipidomic profiling was performed by Metabolon, Inc (Metabolon, Inc, Durham, NC). Instrument variability was determined by calculating the median relative standard deviation (RSD) for the internal standards that were added to each sample prior to injection into the mass spectrometers. Overall process variability was determined by calculating the median RSD for all endogenous metabolites (i.e., non-instrument standards) present in 100% of the Client Matrix samples, which are technical replicates of pooled client samples. Values for instrument and process variability meet Metabolon's acceptance criteria. We have added this in the method section, and have shared the raw data.

6. The authors detail several exciting phenotypes in the Aia4KO mice, the discussion should hint at potentials for further exploration. Some striking findings to follow up on are: 1) The infiltration of macrophages and the activation differences in Aia4KO mice. Others including work from the lab of Amira Klip, have shown that macrophages in the adipocyte provide iron and other macronutrients to the adipocyte which could explain the mitochondrial morphological differences. 2) Another striking phenotype that has been explored is the work of Granneman lab that shows the importance of the macrophages in fat cell turn over and beige adipogenesis.

Response

---Thank you for your interesting question. Macrophage and adipocyte crosstalk is an exciting issue. As you suggested, our mouse model helps analyze how macrophages and adipocytes are activated by each other and how adipocyte turnover is activated, including the homeostasis of mitochondria and iron and macronutrients. In future experiments, we hope to identify which macronutrients affected the mitochondria in Aia4KO adipocytes. In addition, we will search for the factors which induce adipocyte regeneration using our mouse models. As the reviewer suggested, these potential further explorations are described in the Discussion section.

7. The authors are likely aware of the work from the Scherer group demonstrating that tamoxifen induces apoptosis in adipocytes. I am curious if the authors performed a dose response especially in isolated adipocytes to ensure the

apoptotic phenotype isn't due to altered sensitivity to tamoxifen in the control and Aia5KO mice.

Response

---Thank you for the comment. We know Prof. Philip Sherer and previously discussed with him. He may see the method of intraperitoneal administration as important. This is not limited to tamoxifen injection. The response might also vary depending on various factors, even with the same tamoxifen, such as the different syringe sizes, exactly where injection occurred, depth of injection, separate lysis buffer and volume, mouse age, and mouse diet (CD or HFD). We have previously optimized the tamoxifen dose and performed a dose-response, especially in isolated adipocytes, to ensure the apoptotic phenotype is not due to altered sensitivity to tamoxifen in the control and Ai-IRKO and Ai-DKO mice models (Sakaguchi et al., Cell Metab. 2017). We also confirmed that the apparent apoptosis in the adipocyte was not induced in the Control mice. In addition, in this paper, we further created the constitutive knockdown of Alpha4 in adipocytes (A α 4KO) and confirmed the lipodystrophy phenotype, which is consistent with the phenotype of the inducible mouse model (Ai- α 4KO).

REVIEWERS' COMMENTS

Reviewer #1 (Remarks to the Author):

Subject: Phosphatase Protector Alpha4 ($\alpha 4$) is Essential for Adipocyte Maintenance and 2 Mitochondrial Homeostasis through Regulation of Insulin Signaling

Sakaguchi et al. have addressed almost all of my concerns except for one comment below. Overall, the manuscript is much improved and I recommend publication after addressing the one remaining concern:

Supplementary Figure 2. It should be clearly stated that the $\alpha 4$ interaction partners listed in the figure were identified from 1 Control and 2 independent samples for $\alpha 4$ overexpressing cells (which is prone to false positives), and further verification will be needed (either bigger sample size or other method such as Co-IP followed by Western blotting) to ensure they are $\alpha 4$ binding proteins. This is due to the following reasons:

The authors have performed IP and western blotting in Figure 2b (n=3) to confirm the interactions between $\alpha 4$ and PP2A and YBX1. However, for the rest of the proteins in Supplementary Figure 2C (excluding $\alpha 4$), it is not confident that they are $\alpha 4$ binding proteins since the results are from only 1 Control and 2 independent samples, which is prone to false positives due to the small sample size. If two identical aliquots from the same peptide mixture sample are analyzed by LC-MS, it is likely that some proteins will be identified in only one of the aliquots, and even for the proteins identified in both aliquots, there may be large differences in their intensities/peak area. This may artificially lead to high fold changes. In addition, the sample preparation prior to LC-MS analysis also has variations.

Reviewer #2 (Remarks to the Author):

The authors have done an excellent job in responding to critiques.

Reviewer #3 (Remarks to the Author):

The authors have addressed my concerns completely. The results from the luciferase reporter assay are compelling and significantly improved the story. I recommend the manuscript for acceptance with no further revisions and look forward to the follow up work exploring the leptin regulatory axis.

Responses to the REVIEWERS' COMMENTS

REVIEWERS' COMMENTS

Reviewer #1 (Remarks to the Author):

Subject: Phosphatase Protector Alpha4 ($\alpha 4$) is Essential for Adipocyte Maintenance and 2 Mitochondrial Homeostasis through Regulation of Insulin Signaling

Sakaguchi et al. have addressed almost all of my concerns except for one comment below. Overall, the manuscript is much improved and I recommend publication after addressing the one remaining concern:

Response

---We thank the reviewer for the suggestions and positive evaluation.

Supplementary Figure 2. It should be clearly stated that the $\alpha 4$ interaction partners listed in the figure were identified from 1 Control and 2 independent samples for $\alpha 4$ overexpressing cells (which is prone to false positives), and further verification will be needed (either bigger sample size or other method such as Co-IP followed by Western blotting) to ensure they are $\alpha 4$ binding proteins. This is due to the following reasons:

The authors have performed IP and western blotting in Figure 2b (n=3) to confirm the interactions between $\alpha 4$ and PP2A and YBX1. However, for the rest of the proteins in Supplementary Figure 2C (excluding $\alpha 4$), it is not confident that they are $\alpha 4$ binding proteins since the results are from only 1 Control and 2 independent samples, which is prone to false positives due to the small sample size. If two identical aliquots from the same peptide mixture sample are analyzed by LC-MS, it is likely that some proteins will be identified in only one of the aliquots, and even for the proteins identified in both aliquots, there may be large differences in their intensities/peak area. This may artificially lead to high fold changes. In addition, the sample preparation prior to LC-MS analysis also has variations.

Response

--- In response to this critical comment, we have stated in the text and Supplementary Figure 2b. as follows “For the mass spectroscopic proteomic analysis. we compared the Control (n = 1) and $\alpha 4$ overexpressing (n = 2, biological replicates) samples.”

As the reviewer suggested, we have validated the binding of several candidate proteins with $\alpha 4$ using co-IP. The results demonstrate PP6 and Mid2 as the $\alpha 4$ binding proteins. These data are shown in the new Supplementary Figure 2c. The list of the proteomics data of the other candidate proteins was deposited in the source data of Supplementary figure2b.

Reviewer #2 (Remarks to the Author):

The authors have done an excellent job in responding to critiques.

Response

---We thank the comments and the suggestions during the initial reviewing of our manuscript.

Reviewer #3 (Remarks to the Author):

The authors have addressed my concerns completely. The results from the luciferase reporter assay are compelling and significantly improved the story. I recommend the manuscript for acceptance with no further revisions and look forward to the follow up work exploring the leptin regulatory axis.

Response

---We thank the reviewer for the suggestions and final positive evaluation.

--- We thank all reviewers for critical but constructive comments in the overall review process. All comments encouraged us and certainly contributed to upgrading the quality of our manuscript.